

# Methodology for High Quality Mobile Measurement with Focus on Black Carbon and Particle Mass Concentrations

Honey Dawn C. Alas[1], Kay Weinhold[1], Francesca Costabile[2], Antonio Di Ianni[2], Thomas Müller[1], Sascha Pfeifer[1], Luca Di Liberto[2], Jay R. Turner[3], and Alfred Wiedensohler[1]

[1]Leibniz Institute for Tropospheric Research, Permoserstrasse 15, 04318, Leipzig, Germany
[2]Institute of Atmospheric Science and Climate, National Research Council, Via Fosso del Cavaliere, 100 – 00133, Rome, Italy
[3]James McKelvey School of Engineering, Washington University, One Brookings Drive, St. Louis, Missouri, 63130, USA

*Correspondence to*: Honey Dawn C. Alas (alas@tropos.de)

**Abstract.** Measurements of air pollutants such as black carbon (BC) and particle mass concentration in general, using mobile platforms equipped with high time-resolution instruments have gained popularity over the last decade due to its wide range of applicability. Assuring the quality of mobile measurement, data has become more essential particularly, when the personal exposure to pollutants is related to its spatial distribution. In the following, we suggest a methodology to achieve data from mobile measurements of equivalent black carbon (eBC) and $PM_{2.5}$ mass concentrations with high data quality. Besides frequent routine quality assurance measures of the instruments, the methodology includes the following steps. a) Measures to ensure the quality of mobile instruments through repeated collocated measurements using identical instrumentation, b) inclusion of a fixed station along the route containing quality-assured reference instruments and c) sufficiently long and frequent intercomparisons between the mobile and reference instruments to correct the particle number and mass size distributions obtained from mobile measurements. The application of the methodology can provide following results. First, collocated mobile measurements with sets of identical instruments allow identification of undetected malfunctions of the instruments. Second, frequent intercomparisons against the reference instruments will ensure the quality of the mobile measurement data of the eBC mass concentration. Third, the intercomparison data between the mobile optical particle size spectrometer (OPSS) and a reference mobility particle size spectrometer (MPSS) allows for the adjustment of the OPSS particle number size distribution using physical meaningful corrections. Matching the OPSS and MPSS volume particle size distributions is crucial for the determination of $PM_{2.5}$ mass concentration. Using size-resolved complex refractive indices and time-resolved fine mode volume correction factors of the fine particle range, the calculated $PM_{2.5}$ was within 5 % of the reference instruments (MPSS+APSS). However, due to the non-sphericity and an unknown imaginary part of the complex refractive index of supermicrometer particles, a conversion to a volume equivalent diameter yields high uncertainties of the particle mass concentration greater than $PM_{2.5}$. The proposed methodology addresses issues regarding the quality of mobile measurements, especially for health impact studies, validation of modelled spatial distribution, and development of air pollution mitigation strategies.





## 1.  Introduction

Mobile measurements of particulate air pollutants, which are often performed with portable instruments on mobile platforms, have been a trend in air quality monitoring for the past decade. Its fast measurements, ease of use, and relatively low costs have appealed to air quality scientists for its applications on air quality mapping (Ghassoun et al., 2015;Ruths et al., 2014),

exposure studies (Peters et al., 2014;Birmili et al., 2013;Patton et al., 2014;Williams and Knibbs, 2016), and emission factor estimates (Ježek et al., 2015;Karjalainen et al., 2014). In particular, this has been widely used to measure air pollutants which have high spatial and temporal variabilities such as black carbon (or equivalent black carbon – eBC - when measured optically according to Petzold et al. (2013)) or the particulate matter (PM) mass concentration (Peters et al., 2014;Rakowska et al., 2014) and which have significant impacts on both climate and health issues. However, mobile measurements encounter many

challenges. Firstly, due to the mobile, and therefore unstable, nature of the measurements, the small and portable instruments might be sensitive to vibrations and sudden changes in ambient conditions, leading often to false data (Cai et al., 2013;Apte et al., 2011). Secondly, to achieve a concentration representative of the chosen route, measurements have to be done along fixed routes and with high number of repeated runs, requiring considerable time and effort. Thirdly, optical particle size spectrometers (OPSS), which are often preferred for particle mass concentration measurements have a limited size range and

depend highly on the aerosol particle optical properties. The OPSS is a practical choice for measuring size-resolved PM mass concentration for mobile measurements because of its portability, relatively low cost, and near real time measurements. However, the OPSS does not measure the volume equivalent particle diameter. The OPSS measures the optical particle diameter based on scattered light and related to the refractive index of the calibration aerosol. The sizing of the OPSS relies then on the calibration response curve that is based on well-defined particles such as certified polystyrene latex (PSL) spheres

with known refractive index. However, real word scenarios involve particles with varying optical properties which can result to erroneous sizing for the OPSS. Studies (Chien et al., 2016;Rosenberg et al., 2012;Binnig et al., 2007) have suggested ways to improve OPSS data through different calibration and post-processing techniques, mainly to relate optical diameter to aerodynamic diameter, the latter being the more relevant for health studies. However, to calculate mass concentrations of PM, especially for $PM_{2.5}$, the optical diameter has to be related to the volume equivalent diameter and not aerodynamic.

There is a wide variety of portable instruments in the market such as for eBC and PM mass concentration. However, the performance of these instruments (especially the commercialized, low cost sensors) and the differences of the mobile measurement methods employed are still questionable. This may often lead to unreliable, misleading, and non-comparable data, especially for health impact studies, validation of modelled spatial distribution, and development of air pollution mitigation strategies (Castell et al., 2017).

In the last five years, several studies have explored different approaches, both for the measurement procedure and data processing to increase the quality and reliability of mobile measurement data. Birmili et al. (2013) emphasized the importance of regular intercomparisons of the portable devices against reference instruments prior and during mobile measurement campaigns. Cai et al. (2013), suggested the use of a diffusion dryer before an absorption photometer for eBC measurements to





optimize the instrument for personal exposure characterization. More recently, Yu et al. (2016) suggested to measure simultaneously on parallel streets to disentangle the spatial from the temporal variability. Concerning the handling of mobile measurement data, Van den Bossche et al. (2015) and Van Poppel et al. (2013) suggested guidelines for pollutants measured with a bicycle. For example, they suggested exploring different statistics in aggregating data spatially to eliminate the impacts

of single events and perform background normalization to determine the influence of local emissions. However, to our knowledge, there is no comprehensive guidelines for mobile measurements of both eBC and PM mass concentrations with a focus on pedestrian exposure.

The main goal of this article is to propose a methodology for mobile measurements and data processing, which would provide reliable and quality-assured data of spatially resolved eBC and PM mass concentrations. Specifically, we propose

elaborate measurements and post-processing techniques based on meaningful physical assumptions by addressing the limitations of an OPSS.

## 2.    Methodology for quality-assured mobile measurements

This methodology was developed around existing, frequently used, portable instruments as well as highly characterized reference instruments for eBC and PM mass concentration measurements (see Table **1**). It is divided into three parts: 1)

methods to ensure quality of mobile instruments and measurements, 2) methods to ensure the quality of the reference instruments, and 3) mobile measurement strategy and data correction to derive PM mass concentrations from OPSS particle number size distribution measurements.

Table 1 Description of portable and reference instruments.  Additional details are provided in Appendix A.

| Parameter | Instrument | Manufacturer | Principle | Specifications | Time resolution | Platform |
|---|---|---|---|---|---|---|
| **Equivalent black carbon (eBC)** | microAethalometer Model AE51 | AethLabs | Attenuation of light by a particle-loaded filter | $\lambda$ = 880 nm MAC[a] = 12.1 | 1 second | Mobile |
| | Multi-angle absorptions photometer (MAAP) Model 5012 | Thermo Scientific | Absorption of light by a particle-loaded filter | $\lambda$ = 637 nm MAC = 6.6 | 1 minute | Fixed (reference) |
| **Particulate matter (PM)** | Optical particle size spectrometer (OPSS) | TSI | PNSD[b] based on scattering of light of particles | Size range: 0.3-10 $\mu$m | 10 seconds | Mobile |





| | | | | | |
|---|---|---|---|---|---|
| Model 3330 | | | | | |
| Mobility particle size spectrometer (MPSS) | TROPOS | PNSD based on electrical mobility of particles | Size range: 0.01- 0.8 µm | 5 minutes | Fixed (reference) |
| Aerodynamic particle size spectrometer (APSS) Model 3321 | TSI | PNSD based on time of flight of particles | Size range: 0.4-10 µm | 5 minutes | Fixed (reference) |

[a] *MAC =mass absorption coefficient ($m^2\ g^{-1}$)*

[b] *PNSD = particle number size distribution*

The following methodology (Figure 1) is proposed to obtain high quality data from mobile measurements. Note that the items with an asterisk correspond to instrument specific methodology based on target air pollutant:

1. Mobile instrumentation and measurements
   a. Instrument checks and calibration before and after the campaign
   b. Designing of a strategic fixed mobile measurement route
   c. Assuring the quality of mobile measurement data through pre-run protocols or routine
   d. Application of identical mobile instrumentation through collocated mobile measurements
   e. Multiple mobile measurements to achieve representative spatial distribution of concentration
2. Fixed Station Measurements with reference instruments
   a. Instrument calibration and verification before and after the campaign
   b. Selection of a background site which is part of the fixed mobile measurement route
   c. Regular checks and calibration of instruments during campaign (once per week)
   d. Merging of MPSS and APSS size distribution*
3. Sufficiently long and frequent intercomparison periods between the fixed and mobile instruments*
   a. Intercomparison of the eBC mass concentration from mobile platform against the reference absorption photometer*
   b. Adjustment of the OPSS PNSD to the one of the reference MPSS according to aerosol type-dependent
      complex refractive index*
   c. Determination of time-dependent fine mode volume correction factor*
   d. Calculation of $PM_{2.5}$ mass concentration*



**Figure 1. Methodology for highly quality-assured mobile measurements. The task box colours distinguish methods 1, 2, and 3 while the colour of the background separates the period of the application of each method. The green box represents the items in the methodology that are instrument specific. For the purposes of this article, the items within the green box are tailored for eBC and PM measurements from AE51 and OPSS, respectively.**





## 3. Measurements

A mobile measurement experiment was designed to apply the methodology mentioned above as part of an intensive campaign called Carbonaceous Aerosols in Rome and Environs (CARE) in the downtown area of Rome, Italy, in February of 2017. The scientific aim of CARE was to characterize the carbonaceous aerosol in the Mediterranean urban background area of Rome. An overview of this campaign and the first results are presented by Costabile et al. (2017). In the following subsections, the application of the methodology and its results will be demonstrated and discussed.

### 3.1 Mobile instrumentation and measurements

Mobile measurements or "runs" of eBC and PM were carried out using the following instruments which are portable and have high time resolution: an absorption photometer (microAeth® AE51 model, AethLabs, San Francisco, CA USA) and the optical particle size spectrometer (OPSS Model 3330, TSI Inc., Shoreview, MN USA), respectively. Specifications of these instruments are in Table 1. These instruments were placed inside a mobile measurement platform called an aerosol backpack (Figure A 1) together with a GPS and a microcomputer for data acquisition. A detailed description of the aerosol backpack is in Appendix A.

### 3.1.1 Calibration of instruments before and after the campaign

Prior to a measurement campaign, these mobile instruments must undergo a series of quality checks in the laboratory such as leak check, flow check and flow calibrations, unit-to-unit intercomparison, and most importantly, intercomparisons with reference instruments. These should be done to ensure that the mobile instruments are operating correctly and provide high-quality data before deployment. The AE51 units must be compared against a multi-angle absorption photometer (MAAP Model 2012, Thermo, Inc., Waltham, MA USA). The OPSS, on the other hand, must be subjected to a size calibration using a mixture of polystyrene latex (PSL) particles of known sizes. The same procedure should be done after a measurement campaign.

### 3.1.2 Designing of a strategic fixed mobile measurement route

The fixed mobile measurement route must be strategically designed to address the study-specific science question(s). This includes careful consideration of the street topography, the length of the route, and the time it takes to complete. The route should include different microenvironments to capture the spatial variability of the pollutant concentration. In deciding on the length of the route and the duration of the run (a "run" is one completion of the route), the operating time of the instruments and rest time for the runners should be considered. If multiple runs are done within one day, the charging time of the instruments should be considered as well. For this study, the fixed route (Figure 2) was approximately nine (9) km long and took 2.5 hours to complete. This route covered different microenvironments to simulate varying exposure scenarios: a park area, roadside, intersections, street canyons, street cross sections, residential and commercial areas, and a gated garden.



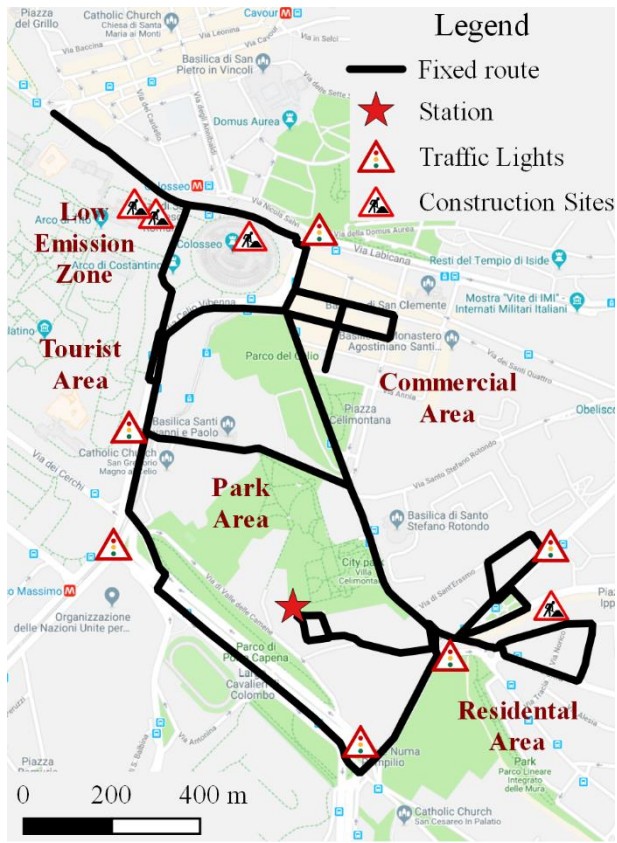

**Figure 2. Illustration of the fixed mobile measurement route with labels of the different microenvironments.**

### 3.1.3    Assuring the quality of mobile measurement data through pre-run protocols or routine

Each mobile measurement period includes a pre-run routine that includes the following: 1) replacing the filter of the AE51 with a new one to avoid filter saturations, 2) checking for leaks within the systems by placing a total filter on the inlet, 3) giving ample time for the instruments to warm up (depending on the instruments used), 4) measuring the total flow of the system, and 5) synchronizing the time of the two microcomputers or data loggers of each backpack. Additionally, if the pre-run routine is done indoors, then once stepping outside, the GPS should be given enough time to find satellites to get accurate location data before starting the run.

### 3.1.4    Application of identical mobile instrumentation through collocated mobile measurements

Deploying a single aerosol backpack may be more practical, however, there is a risk of completing a sampling period without knowing if the data is valid, especially in the absence of real-time data viewing. This is highly like for portable instruments which are sensitive to vibrations, sudden movements and changes in its immediate environment. Here we demonstrate the advantage of performing collocated measurements with two aerosol backpacks containing identical instrumentation carried by two "runners".



Having two, identical aerosol backpacks opens several possibilities for mobile measurements. One example is to determine the spatial variability of pollutants by deploying each aerosol backpack in parallel streets at the same time (Yu et al., 2016). This way, the spatial aspect can be separated from the temporal. One can also perform runs along the same route but 30-minutes apart or so. This would result to high number of data points leading to increased representativeness of the

overall spatial average. These two examples are best applied, when data can be viewed and checked in real time so the quality is not compromised. Another way is to target quality assurance by doing side by side parallel runs or collocated measurements as done here. This allows for a constant quality check of the mobile instruments along the whole route, since live visualization of data is not available, particularly in locations without reference instruments to compare with. During this campaign, mobile measurement data can only be checked after the run, and consequently, errors can only be noticed during the post-processing

of the data. For example, during the early stages of the campaign, analysis of the collocated measurements revealed that one AE51 was underestimating eBC mass concentrations. This was not flagged by the instrument, but because another AE51 was in operation, the error was identified and corrected immediately. Similarly, towards the end of the campaign, the sheath flow of one of the OPSS started to increase which resulted to an underestimation of the particle number concentration (PNC) across all size bins. This was not flagged by the instrument and was only noticed when compared against the other OPSS. By

comparing the data gathered from the two aerosol backpacks post-run, errors in the data were easily noticed, investigated and corrected, especially errors that were not flagged by the instruments (i.e. sheath flow drift, time shift, etc.).

To demonstrate the advantage of this approach, an example of a time series from one run is shown in Figure 3. The top most panel shows the time series of eBC mass concentrations from the two AE51s in 10s resolution (median). While the last two panels show the time series of PNC at 0.417 µm and 2.406 µm, respectively, measured by the two OPSSs. The scatter

plots on the right of each time series shows the correlation between the two corresponding instruments. The collocated measurements show the performance of the mobile instruments throughout the whole route which are in good agreement with each other. The peaks in the data represent the parts of the route which are closer to sources such as street sides and intersections. The correlation analyses show that the unit-to-unit variability of the two AE51s were within 10 % and within 5 % for the two channels shown here for the OPSS units. The small discrepancies between the mobile instruments are attributed

to instrument uncertainty. Large differences, on the other hand, were investigated further to determine if it is related to sources or technical malfunctions. For example, in one of the runs, the pump of AE51 (2) was not working properly making the sample flow 50 % lower than desired. A comparison with AE51 (1) showed that the eBC mass concentration measured by AE51 (2) was underestimated. This was also verified through intercomparison against the MAAP. Hence, the eBC measurements from AE51 (2) during this run was removed from the dataset. Similar cases were filtered out during the data selection process.






**Figure 3. 10-s resolution time series of eBC (a) and PNC from two channels (c and e) of the OPSS from one parallel run. The panels on the right show the correlation between the AE51s (b) and OPSSs (d and f) using standard major axis regression to account for the error on both axes.**

5    ### 3.1.5    Multiple mobile measurements to achieve representative spatial distribution of concentration

Pollutants such as BC and $PM_{2.5}$ can be highly variable in space and time, especially at ground level in urban areas. Determining the spatial distribution of these pollutants that is representative of the study area would require a high number of repeated measurements over different periods to minimize the impacts of single events such as a passing of a heavy-duty vehicle or the presence of a temporary construction area. Here, we demonstrate the advantage of performing a high number of



runs along a fixed route. Runs were done three times a day (morning rush hour, midday, and evening rush hour), weekdays and weekends for the whole CARE campaign. This resulted in a total of 77 runs, distributed across conditions as shown in Table **2**.

Table 2 Summary of mobile measurement runs during the CARE campaign

|  | # of days | # of runs |
| --- | --- | --- |
| **overall** | 28 | 77 |
| **weekdays** | 20 | 54 |
| **weekends** | 8 | 23 |
| **mornings** | 26 | 26 |
| **middays** | 27 | 27 |
| **evenings** | 24 | 24 |

Minimizing the impacts of single events can also be achieved by choosing an appropriate averaging method for the data collected within a run. Van den Bossche et al. (2015) have investigated the results of using different averaging methods and concluded that minimizing the impacts of single events is best achieved by using the trimmed-mean. Others (Peters et al., 2013;Brantley et al., 2014;Alas et al., 2018) have argued that the use of median is more robust to bias due to single events. The selection of an averaging method depends on the scientific question for a particular study. Here, the nonoverlapping interval 10-second median was used in order to maintain a high spatial resolution.

**Convergence Analysis**

As previously stated, spatial data are highly variable because of many factors and a single or a few runs will not provide a representative estimate of the spatial distribution. Data experiments to determine the number of runs necessary to achieve a representative estimate of pollutant concentration based on mobile measurements have been introduced by Peters et al. (2013), Van Poppel et al. (2013) and Van den Bossche et al. (2015). This is done by taking the cumulative average of randomly selected runs along a specific part of the route with high number of iterations to ensure a high number of possible combinations. Convergence is achieved when the desired metric (e.g. median concentration) has stabilized to an asymptotic value within an acceptable tolerance. The criteria for convergence depends on the science question and goals to be achieved. Previous experiments have used convergence criteria based on required accuracy of $PM_{2.5}$ mass concentrations obtained from air quality monitoring stations (25 %) and indicative measurements (50 %) and the latter was used in this study for demonstration. Figure 4 shows the results of the convergence analysis performed on a street canyon and a park area as examples. For the park, since there are no direct sources (it is inaccessible to vehicles) the data converged faster at 56 runs while in the street canyon which has higher variability because of traffic emissions, convergence was achieved at 67 runs. This shows that for routes containing regions with higher concentration variability, mobile measurements must be conducted with





higher number of repetitions. It must be emphasized, however, that one must consider a threshold that accounts for the natural variability of the pollutant concentration.

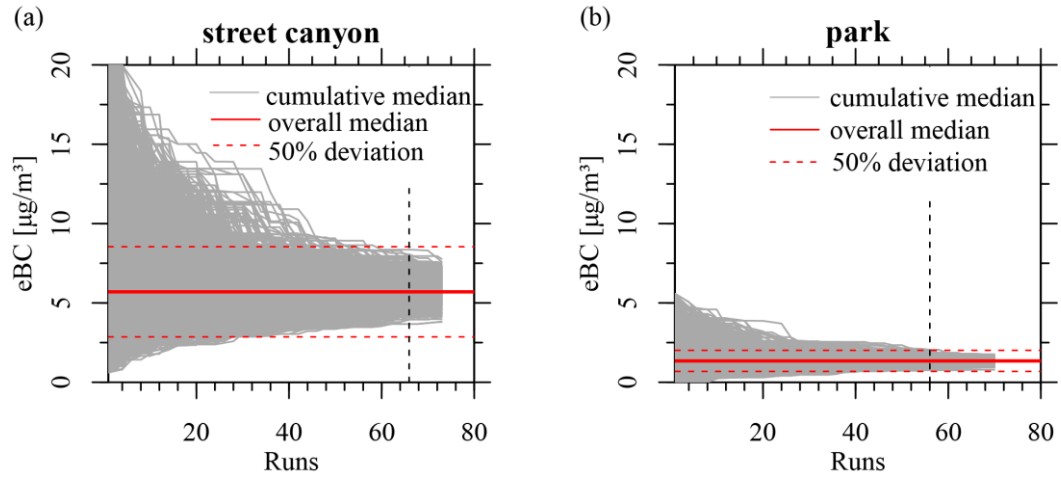

**Figure 4. Convergence analysis performed on data points in the (a) street canyon and (b) park area. The vertical dashed line marks the number of runs where convergence was reached.**

**Spatial Averaging**

   The data points resulting from mobile measurements do not exactly fall on the same point in space and time. The time-resolution of the instruments as well as the uncertainty of the GPS contribute to this. Figure 5 shows the cloud of data points (purple dots) acquired from all runs. Therefore, to obtain the overall spatial distribution, the data points has to be spatially aggregated. Prior to spatial aggregation, the data cloud has to be cleaned by removing data points that is not part of the route (e.g. detours, inaccurate GPS points). The spatial aggregation method used in this study was that of Alas et al. (2018). Briefly, the pre-determined route was created with equidistant points (0.0002°, or ~23 m; green dots in Figure 5a). The distance between the two points depends on the desired spatial resolution. The data points are projected on a map as a data cloud. These points, particularly the ones that are not on the street due to GPS uncertainty, are snapped back onto the nearest street using a point-snapping algorithm.  The pre-determined route was used as centre points for spatial aggregation. The median of all the data points falling within a user-defined radius around each centre point (in this study 0.0005°, or ~56 m) is calculated, resulting in a moving circular median. Figure 5b shows the result of the spatial aggregation with the colours indicating the pollutant concentration, in this case, the eBC mass concentration.




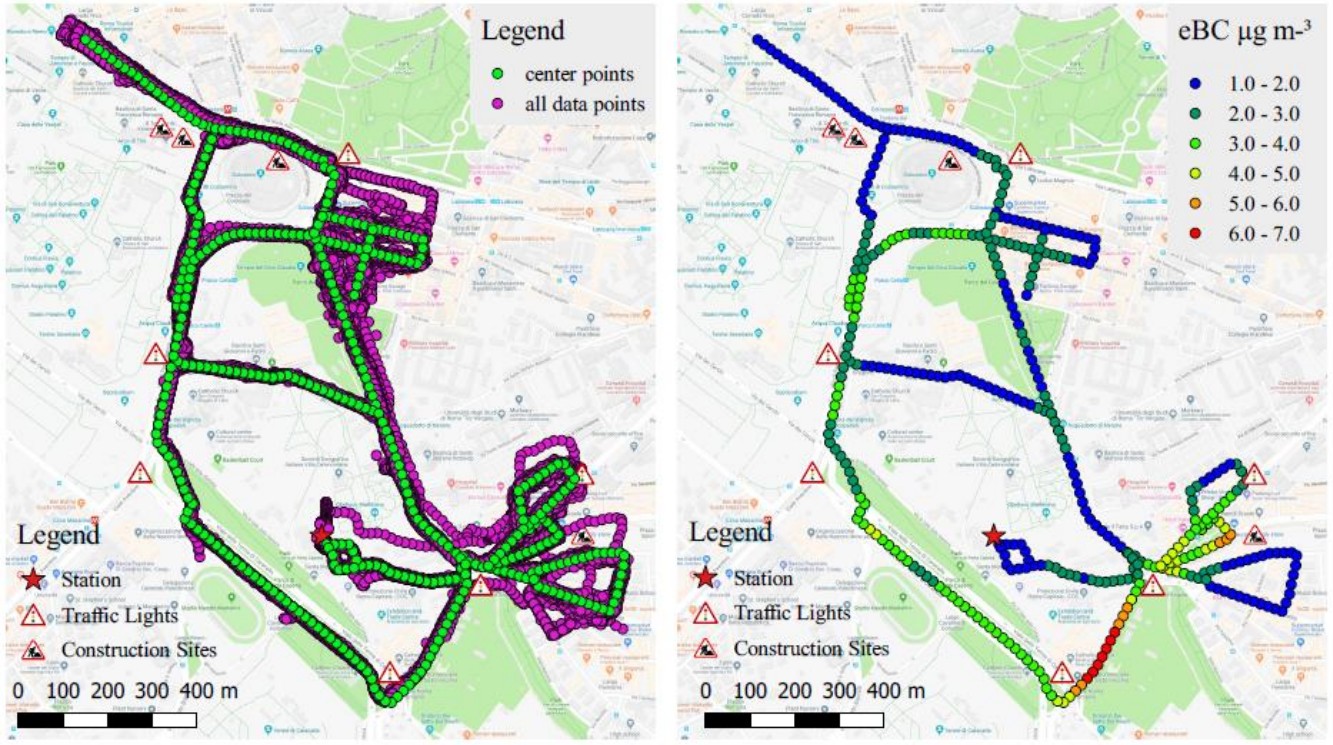

**Figure 5. Illustration of the spatial averaging performed on the mobile measurement data with (a) showing all the data points in purple dots and the centre points for averaging in green and (b) the spatial average of eBC mass concentrations for the entire campaign.**

## 3.2 Fixed Station Measurements

Table **1** describes the reference instruments and brief descriptions of the fixed station, as well as the operating principle of each reference instrument are in Appendix B. Detailed information can be found in Costabile et al. (2017)

### 3.2.1 Calibration and verification of instruments before and after a measurement campaign

Prior to a measurement campaign, the reference instruments must undergo checks and calibration in the laboratory to ensure proper operation. The following paragraphs briefly describe the checks and calibration procedures for each of the reference instruments.

The standard procedures for assuring the quality of the MPSS involves checking for leaks by attaching a total filter on the inlet, as well as aerosol and sheath air flow checks. Furthermore, the MPSS must undergo sizing accuracy check by using standard PSL particles. A high voltage calibration should also be done to ensure correct determination of electrical mobility. Preferably, the PNSD and PNC of the MPSS must be compared against the reference TROPOS MPSS by, for example, participating in regular international workshops done at the World Calibration Center for Aerosol Physics (WCCAP). The complete calibration procedure is described in Wiedensohler et al. (2018).





For the APSS, the flow rates must be checked and adjusted from deviations since the sizing accuracy and counting are highly dependent on them. Furthermore, a sizing calibration using a mixture of PSL particles of known sizes should be performed. The calibration procedures are discussed in detail in Pfeifer et al. (2016).

Similarly, the MAAP should undergo quality assurance by performing sensor calibrations outlined in the instruction manual, particularly the flow calibration. It has been reported that the unit-to-unit variability of the MAAP (expressed as coefficient of variation) is reduced to 3 % from 11 % after flow calibration (Müller et al., 2011). As there is no standard for BC measurements, the MAAP should undergo regular unit-to-unit intercomparisons or compared against other BC measurement methods (thermal/optical methods, etc.). Full details of the performance of the MAAP and how it compares to other absorption photometers can be found in Petzold and Schönlinner (2004) and Müller et al. (2011).

### 3.2.2 Selection of background site which is part of the fixed mobile measurement route

The fixed station containing reference instruments is crucial for the quality assurance of the mobile instruments and also for the determination of $PM_{2.5}$ mass concentration derived from the PNSD of the OPSS mobile measurements. Therefore, the selection of the fixed measurement site should be taken with care. Preferably, the fixed station should be in a background site where there is less variability of the pollutant concentrations. The decision on the reference site location is a balance between scientific aims and availability of space. Since it was conducted in an urban area of Rome, the fixed station was situated in an urban background area - inside a government-owned garden that is inaccessible to most vehicles.

### 3.2.3 Regular checks and calibration of instruments during the measurement campaign

Regular checks and calibration of the instruments are necessary to assure the quality of the data gathered at the fixed station during the campaign. This includes the following: leak check of the station's sampling manifold (by placing a total filter on the main inlet of the station), flow and leak checks of each instrument inside the station, sizing calibration of the MPSS and APSS using PSL particles, and high voltage calibration of the MPSS. For this study, these procedures were done every week.

### 3.2.4 Merging of MPSS and APSS particle number size distributions

Deriving PM from the merged PNSDs of MPSS and APSS leads to high time resolution data of PM, which is an advantage over filter-based measurements which uses gravimetric analyses to obtain PM mass concentrations. The following paragraph briefly describes the merging process, which can be found in detail in Costabile et al. (2017). First, the PNSD measured by the APSS, based on aerodynamic particle diameters ($D_{p,aer}$), was converted to a  PNSD based on the volume equivalent particle diameters ($D_{p,voleq}$). The conversion assumed spherical particles (shape factor = 1) and a size-dependent particle density (1.6 to 2 g cm$^{-3}$). It was assumed that the particles in the fine particle range were spherical or rather compact and that the mobility particle diameter is equal to the volume equivalent particle diameter. Figure 6**Error! Reference source not found.** shows an example of the PNSDs measured by MPSS and APSS, including the indication of their overlapping size





range from $D_{p,voleq}$ 0.475 to 0.830 µm. For this size range, a combined PNSD was obtained by fitting a power-law function to the overlapping size range of the PNSDs of the APSS and MPSS. The two size distributions were merged by varying the PNSD expressed as dN/(dlog(D)) after Khlystov et al.'s (2004) results. The fitting was constrained by an iterative procedure based on the minimization of the relative square difference between the PNSDs. The final PNSD covers the range from 0.01-10 µm

5     volume equivalent particle diameter. The merged particle volume size distribution (PVSD) can be converted to particle mass size distribution (PMSD) by multiplying the former with the density of the particle of a certain size. The results of this procedure serve as a size distribution-derived PM reference with high time resolution which is necessary for the intercomparison against the mobile instruments.

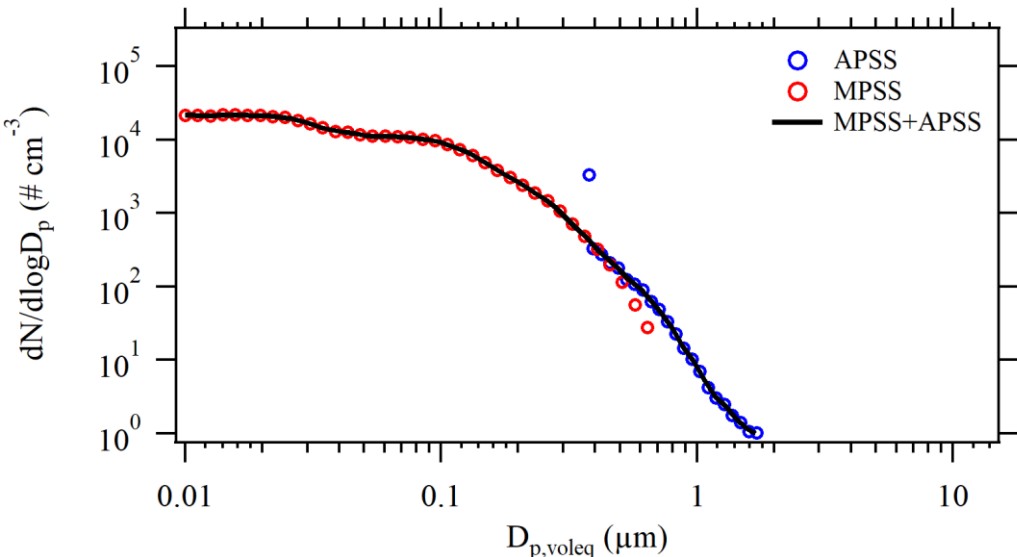

**Figure 6. Illustration of the procedure of merging the PNSDs of MPSS and APSS.**

      However, there are other options for the traceability of the mass concentration measurements. These options include, online mass monitors, filter-based measurements, and chemical analyses. In this study, as reported by Costabile et al. (2017), the PM$_1$ derived from the MPSS size distribution was compared ($r^2 = 0.98$, y=0.97x) against the PM$_1$ reconstructed from the

15     MAAP and the Aerodyne aerosol chemical speciation monitor (ACSM). The PM$_{2.5}$ and PM$_{10}$ derived from this procedure were compared against the ones measured by a beta attenuation monitor (BAM) from an urban background station 3 km away. The PM$_{2.5}$ measurements compared well with each other ($r^2 = 0.86$, $p < 0.001$), with no significant difference in the 24h mass concentration. The agreement for PM$_{10}$ is lower, probably because of dust events, local re-suspended dust as well as differences of sources between the two stations. More information regarding the correlation of the fixed instruments against other methods

20     employed either in the same station or in another nearby monitoring station are presented in the Supplementary Material from Costabile et al. (2017).





### 3.3 Sufficiently long and frequent intercomparison periods between the mobile and fixed instruments

To further ensure the quality of the mobile measurements, sufficiently long and frequent intercomparison against the reference instruments in the middle of a run is recommended. Here, the runners stop by the fixed station for 30 minutes in the middle of each run for intercomparisons against the reference instruments. During the weekends the garden was closed and the reference station was inaccessible. For these times, the mobile measurements along a park area 460 m north of the aerosol container were used as a proxy for intercomparison against the reference instruments. This park area and the reference station were separated by a large green park inaccessible to vehicles. The results of these intercomparisons are discussed in the following paragraphs.

### 3.3.1 Intercomparison of the eBC mass concentrations from the mobile platform against the reference absorption photometer

For the eBC mass concentration measurements, the AE51 data was averaged per minute to compare with the MAAP. The AE51 data compared were within 5 % of the MAAP during the intercomparison periods (Figure 7). This increases our confidence that the measured eBC mass concentration are reliable for the entire route.

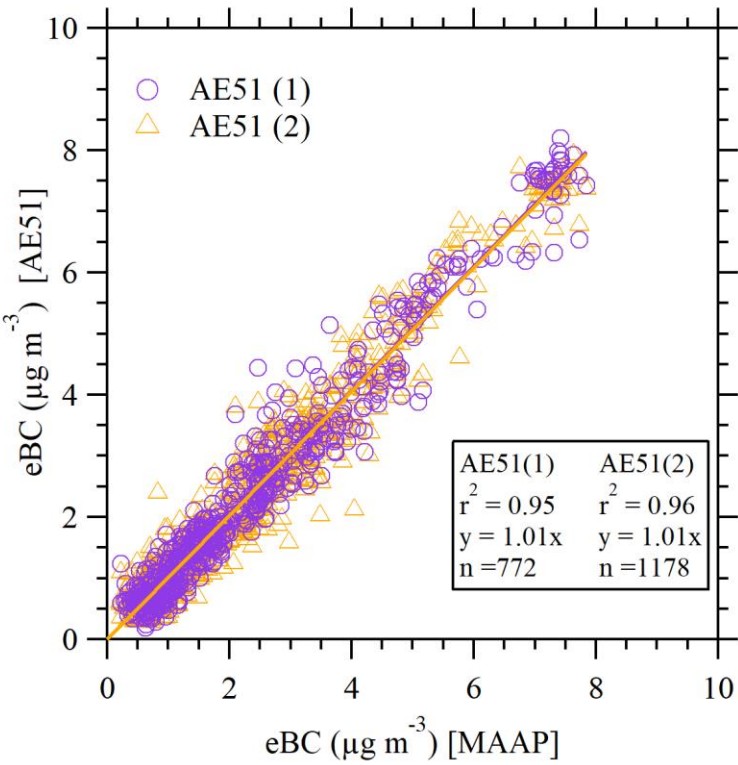

**Figure 7. Correlation between the two AE51s (in 1-minute average) aboard the aerosol backpacks against the reference absorption photometer MAAP during the intercomparison periods.**





### 3.3.2 Adjustment of the OPSS PNSD to the reference MPSS PNSD according to complex refractive index (ñ)

As mentioned, the OPSS has been calibrated by the manufacturer with certified PSL particles of known sizes. These particles are spherical and non-absorbing with a complex refractive index (ñ) of 1.59 – 0.0i. This leads to inaccuracies of the optical PNSD from the OPSS, when used to measure ambient aerosol which has particles of various shapes, sizes, and ñ.

Therefore, to derive PM mass concentrations from an OPSS, the optical PNSD must be adjusted with a ñ typical for the aerosol type of the study area. To achieve this, the OPSS must be compared consistently and frequently to a reference MPSS, which does not depend on particle optical properties such as ñ. Converting the optical particle number/volume size distribution of the OPSS by an aerosol type-dependent ñ, to an equivalent particle number/volume size distribution can yield reasonable results for the submicrometer size range compared to the MPSS derived size distributions, assuming compact and spherical-like

particles. Here, we demonstrate step-by-step the complex refractive index correction using the Mie theory on the OPSS data with the MPSS PNSD as reference using one of the 30-minute intercomparison periods from a single run as an example.

Figure 8a shows the deviation between the particle volume size distributions (PVSD) of the OPSS from the MPSS before any correction. This deviation is due to the different diameters (optical for the OPSS and mobility for the MPSS) measured by the two instruments, as well as the inaccuracy of the OPSS due to ñ based on PSL. When the correction is applied

using the Mie theory, the optical diameters have been converted to geometric mean volume equivalent diameter. Figure 8b shows the effect of adjusting the real part ($\tilde{n}_{re}$) of the ñ to values typical of urban areas (1.51 – 0.0i) but keeping the imaginary part ($\tilde{n}_{im}$) zero. This yielded to an OPSS PVSD in the submicrometer range that is reasonable when compared to the PVSD of the MPSS. Since the urban aerosol contains absorbing particles (BC, mineral dust, etc.), the imaginary part should not be zero. However, Figure 8c shows that increasing the $\tilde{n}_{im}$ results to artificial overestimation of the supermicrometer PVSD. This is due

to two reasons: (i) the optical particle diameter of supermicrometer particles is sensitive to slight changes of the $\tilde{n}_{im}$ of the refractive index, in the range from 0.0 to 0.01; and (ii) with increasing particle size of atmospheric aerosol particles, their shape also becomes more non-spherical, leading to unpredictable phase functions of the particle light scattering inside of the optics of the OPSS. Even for a latex-calibrated OPSS, this effect of non-spherical particles is not correctable since the refractive index correction by Mie theory is only possible for spherical particles.

In addition, the two instruments being compared have different operating principles (for the coarse mode OPSS and APSS). It must be noted as well that the intercomparisons were such that the aerosol backpack and the reference systems are not in the same inlet and at different heights (aerosol backpack's inlet is ~ 1.5 m above the ground while the fixed station inlet is ~3.5 meters above the ground) which could significantly influence the coarse mode concentrations (higher coarse mode particle concentrations are observed closer to the ground due to, for example, resuspension of particles).

Further attempts to correct the supermicrometer range by using a different ñ did not result to any significant improvement of the OPSS PVSD as shown in Figure 8d. However, using size-resolved complex refractive indices (if they are known) is ideal as it might help in decreasing the difference between the PVSD of the OPSS and the MPSS. The results above show that the refractive index correction using Mie theory is not applicable to the supermicrometer size range.

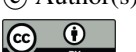



From the result of the data experiments discussed above, the following were assumed in this study: $\tilde{n} = 1.51 - 0.007i$ and $\tilde{n} = 1.56 - 0.005i$ for particle dimeters 0.3-0.8 µm and diameters larger than 0.8 µm, respectively. This yielded corrected $D_{p,voleq}$ for the OPSS ranging from 0.46-12.02 µm after neglecting the first channel of the OPSS due to inaccuracy.

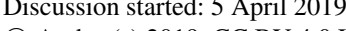

**Figure 8.** Demonstration of the complex refractive index correction done on the OPSS size distribution. In (a), the x-axis for the OPSS is optical diameter ($D_{p,opt}$). In the succeeding panels (b-d), the OPSS PVSD has been corrected and the x-axes are geometric mean volume equivalent diameters ($D_{p,voleq}$).




### 3.3.3 Determination of time-dependent fine mode volume correction factors (CF$_{f,vol}$)

Another major limitation of the OPSS is that it measures only a fraction the PSD, in this case, only particles with optical diameter ranging from 0.3 µm to 10 µm. In Figure 9, the comparison of the PVSD from the OPSS and MPSS shows that the OPSS misses the fine mode volume peak (~ 0.3 µm). This means that PM calculations from the OPSS PNSDs may lead to significant underestimation of the mass concentrations. In order to address this, relatively long and frequent intercomparison periods between the OPSS and the MPSS should be performed for each run. From these intercomparison periods, a correction factor (CF$_{f,vol}$) for the fine mode of the OPSS can be calculated based on its ratio with the MPSS fine mode (Eq.1). This correction must be done on each run because of the dynamic nature of the fine mode volume peak diameter (peak diameter from here on) and on the volume size distribution itself.

$$CF_{f,vol} = \frac{MPSS\ fine\ mode\ [\,0.001\ \mu m - 0.8\ \mu m\,]}{OPSS\ fine\ mode\ [0.4\ \mu m - 0.8\ \mu m]}$$
Eq. 1

Figure 9**Error! Reference source not found.** shows the VSDs of the MPSS and OPSS for two examples (high and low CF$_{f,vol}$). Fine mode coverage by the OPSS varies with changes in the PVSD. Using the peak diameter as a proxy for these changes, as the peak shifts to the right the OPSS PVSD covers a larger portion of the fine mode (lower CF$_{f,vol}$) as compared to when the peak is located more to the left (higher CF$_{f,vol}$). This variability of the peak diameter measured by the MPSS depends on the local sources in urban areas. Since these local sources have a diurnal variability, so does the peak diameter.

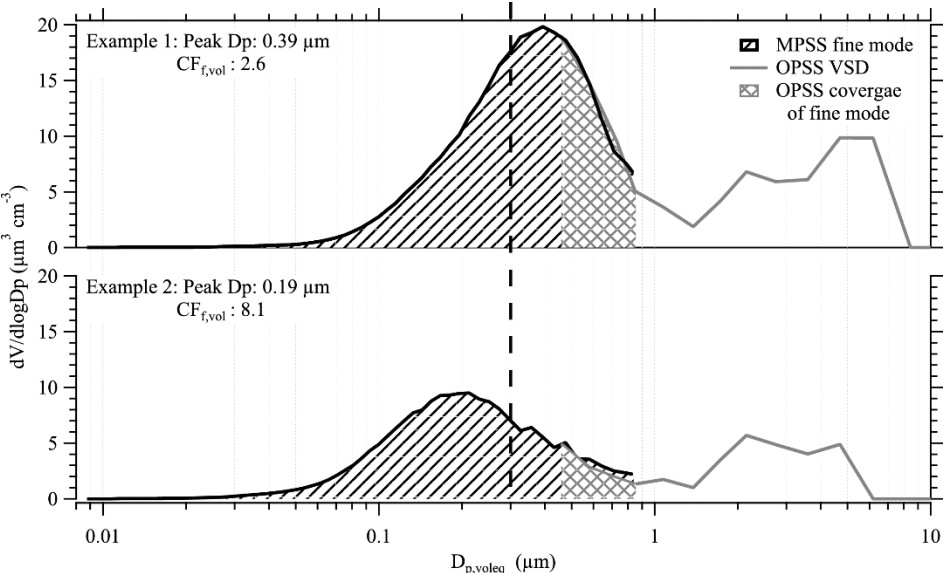

**Figure 9. Dependence of the OPSS coverage of the fine mode on the variability of the fine mode peak diameter.**

Figure 10 summarizes the variability of the peak diameter from the MPSS (during intercomparison periods) along with the corresponding CF$_{f,vol}$ for each run. There exists no significant pattern among the CF$_{f,vol}$ and hence it is necessary to correct each run individually. Since there are no other fixed monitoring stations with an MPSS along the rest of the route, the same correction factor was used to correct the fine mode of each data point along the whole route of one run. This assumption comes





with limitations as it doesn't account for the likely differences of the aerosol sources along the entire route. A data experiment was performed comparing PVSDs obtained at an urban background station and at a roadside station in the city of Dresden, Germany for the whole month of February 2017. For each site, effective correction factors ($CF_{f,vol}$) were calculated for each hour between 6 AM and 9 PM using the fraction covered by the OPSS as a proxy for the OPSS size distribution. For background

station $CF_{f,vol} < 2$, which represented ~50 % of the hourly data, there was excellent agreement between the paired background and roadside $CF_{f,vol}$ values with 3 % mean bias and narrow variability ($1\sigma = 5$ %). For higher $CF_{f,vol}$ values the bias increased with increasing $CF_{f,vol}$ and approached 20% for $CF_{f,vol} > 3$ (background station having higher values and the variability also increased ($1\sigma = 11$ %). While caution must be used in extrapolating the Dresden data to other locations and conditions, these results provide a context for understanding the limitations when using a correction factor derived at a single location to

represent the behaviour along the entire route. For most cases, the impacts of location-dependent $CF_{f,vol}$ values will be damped through the use of repeat runs although some concentration bias might remain.

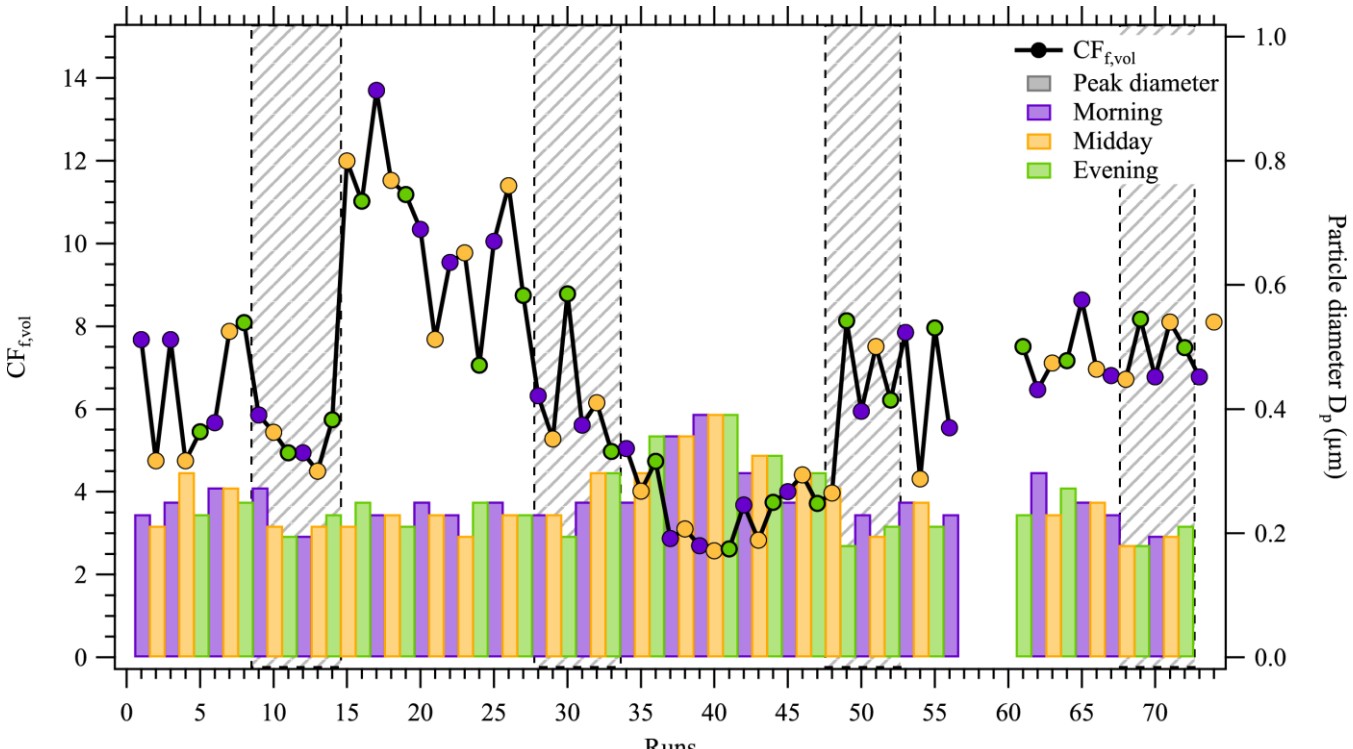

**Figure 10. Variability of the peak diameter and CF$_{f,vol}$ colour coded according to time of day.**

### 3.3.4 Calculation of PM$_{2.5}$ mass concentration from the OPSS number size distribution

Once the $CF_{f,vol}$ have been determined for each run, PM mass concentrations can be calculated. The approach applied in this study was to correct PM$_1$ of the OPSS and add this absolute value to PM$_{1-2.5}$ and PM$_{1-10}$ to get PM$_{2.5}$ and PM$_{10}$, respectively. First, the OPSS PVSD was converted to a mass size distribution using a size-resolve particle density. Secondly,



since the PM threshold is defined by $D_{p,aer}$, the equivalence of $PM_1$ and $PM_{2.5}$ to $D_{p,voleq}$, which serve as the new limits of integration, were determined following Eq. (2) which assumes spherical particles (shape factor = 1).

$$D_{p,voleq} = D_{p,aer}\sqrt{\frac{\rho_o}{\rho_p}} \qquad \text{Eq. 2}$$

where $\rho_o$ is the reference density (1 g cm$^{-3}$) and $\rho_p$ is size dependent particle density (1.6 and 2 g cm$^{-3}$ for $PM_1$ and $PM_{2.5}$ and

$PM_{10}$, respectively, Costabile et al. (2017) Supplementary Material). Therefore, the particle mass size distributions were integrated from $D_{p,voleq}$ 0.4-0.8 µm, 0.8-1.7 µm, 0.8-7.8 µm for $PM_1$, $PM_{1-2.5}$, and $PM_{1-10}$, respectively. Linear interpolation was used to align the OPSS bin endpoints with these cut-off values. The $PM_1$ mass concentrations were corrected using Eq. (3) and $PM_{2.5}$ and $PM_{10}$ were calculated using Eq's. (4) and (5), respectively.

$$corrected\ PM_1 = CF_{f,vol} \times PM_1 \qquad \text{Eq. 3}$$

$$PM_{2.5} = corrected\ PM_1 + PM[0.8\ \mu m - 1.7\ \mu m] \qquad \text{Eq. 4}$$

$$PM_{10} = corrected\ PM_1 + PM[0.8\ \mu m - 7.8\ \mu m] \qquad \text{Eq. 5}$$

Figure 11 shows that the median $PM_{2.5}$ derived from the OPSS compares well with the one derived from the merged MPSS + APSS size distribution ($r^2 = 0.98$, y = 1.002x) during each intercomparison period. The resulting spatial distribution

(Figure 12) yielded reasonable results as well.

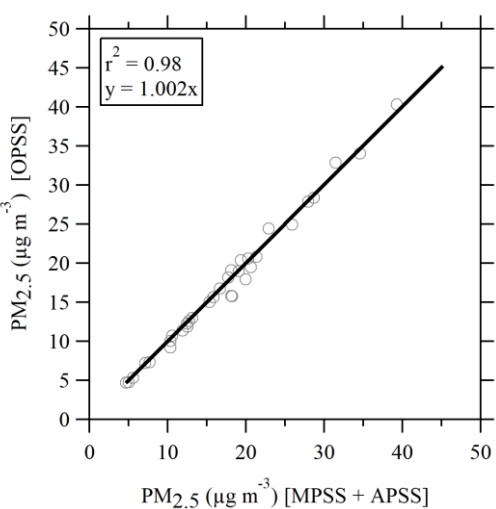

**Figure 11. Correlation between the PM$_{2.5}$ derived from the OPSS and MPSS+APSS.**





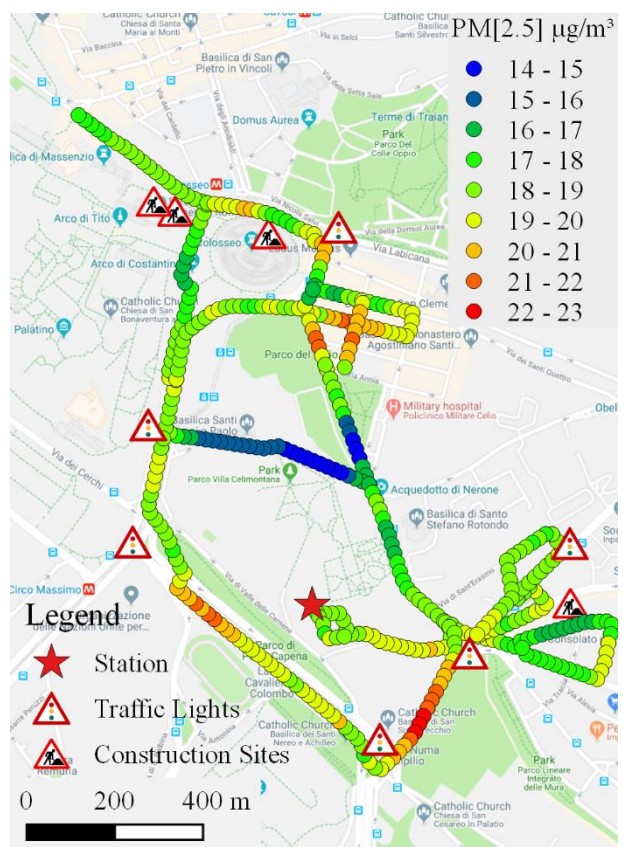

**Figure 12. Overall spatial distribution of PM$_{2.5}$ along the fixed route during the CARE campaign. Map source: Google Maps**

Furthermore, the impact of each correction procedure on the PM$_{2.5}$ value was investigated. Figure 13 shows that the deviation from the reference PM$_{2.5}$ decreases with the successive application of each step in the correction procedure. Without

5  performing any correction on the OPSS data, PM$_{2.5}$ is underestimated by 74 %. When a refractive index correction is performed and a mean CF$_{f,vol}$ is used, PM$_{2.5}$ values are 34 % higher than the reference. Finally, the deviation from the reference is significantly minimized when a size-resolved refractive index correction is used and unique CF$_{f,vol}$ is applied for each run.





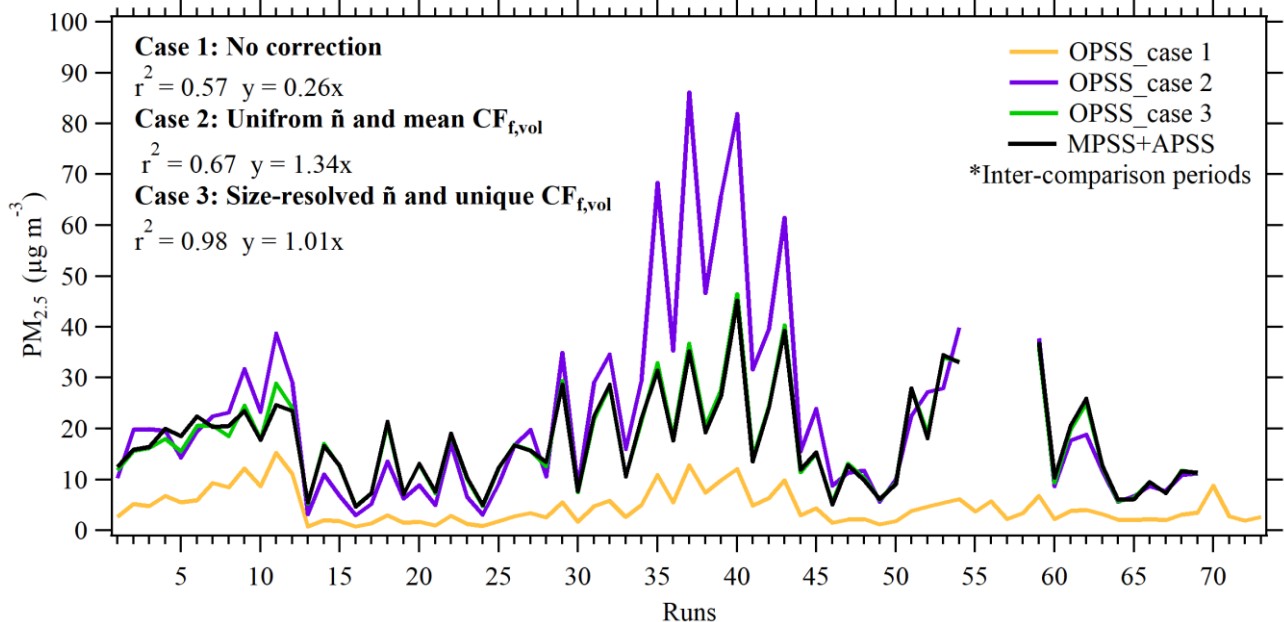

**Figure 13. Illustration of the impact of data correction on the uncertainties of the PM$_{2.5}$ from OPSS.**

The procedure outlined here to correct the size distributions (PNSD and PVSD) from the OPSS worked well with PM$_{2.5}$ but yields large uncertainties (>50 %) when higher size fractions are calculated such as PM$_{10}$. As mentioned, this is attributed to the increasing irregularity in shape of larger particles, making the refractive index correction method based on the Mie theory ineffective. Therefore, other methods should be explored to correct the supermicrometer range of the OPSS size distribution.

## 4 Conclusions

A methodology to assure high quality mobile measurement data of eBC and PM$_{2.5}$ mass concentrations was introduced and validated during an intensive field study in Rome, Italy, February 2017. The concept includes three main aspects: a) The quality assurance of the mobile instruments and strategic design of the mobile measurements, b) the quality assurance of reference instruments, including a fixed station in the mobile measurement route, and c) sufficiently long and frequent intercomparison periods between the mobile and reference instruments as a basis for correcting the OPSS particle number size distributions. The concept proved effective in assuring the quality of the data from the mobile measurements. Performing collocated runs allowed for constant unit-to-unit intercomparison between mobile instruments leading to the detection of errors not flagged by the instruments operating in isolation. Fixed station measurements were used to frequently check the performance of the mobile instruments and to derive PM mass concentrations by referencing the OPSS to a mobility size spectrometer. This study also demonstrated that a correction of the OPSS data, using aerosol type-dependent complex refractive indices and unique fine



mode volume correction factors, significantly improved calculated PM$_{2.5}$ mass concentrations. However, large uncertainties were observed for the PM$_{10}$ mass concentration.

Data of this quality can be beneficial to increase the accuracy of exposure estimates as well as in validation of microscale models. Moreover, the high spatial resolution data can prove valuable for policy-makers and urban planners in developing
strategies to mitigate air pollution.

**Author contributions:** Honey Dawn Alas, Kay Weinhold, Francesca Costabile, Antonio di Ianni, and Alfred Wiedensohler conceived and designed the mobile measurements. Thomas Müller and Sascha Pfeifer designed the aerosol backpack, wrote data acquisition and processing codes. Luca di Liberto and Kay Weinhold operated, maintained, and did the regular calibrations
and checks of the reference instruments at the fixed station. Honey Dawn Alas, Kay Weinhold, Francesca Costabile, Antonio di Ianni (together with others mentioned in the Acknowledgements) did the mobile measurements. Honey Dawn Alas wrote the paper with contributions from Kay Weinhold, Francesca Costabile, Jay Turner, and Alfred Wiedensohler. All the authors contributed with comments and suggestions on the manuscript, and approved it.

**Competing interests:** The authors declare no competing interests.

**Data Availability:** All the data presented in this study is available from the authors upon request.

**Acknowledgements**

We thank all the research groups who participated in the CARE experiment, and financially supported it. Also, special thanks
to G.P.Gobbi (ISAC, CNR), G.Montanari and S.Simoni (Environmental Sustainability Department of Rome), and S. Ubertini (University La Tuscia of Viterbo) for their support. We would also like to thank the Saxonian "Staatliche Betriebsgesellschaft für Umwelt und Landwirtschaft" (BfUL) and the Saxonian "Landesamt für Umwelt, Landwirtschaft und Geologie" (LfULG) for the Dresden, Germany data. Finally, we are truly grateful to Mario Benincasa, Riccardo Biondi, Giacomo Bigazzi, Francesca Barnaba, and Spartaco Ciampichetti for their help with the mobile measurements.

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

©c Author(s) 2019. CC BY 4.0 License.



Binnig, J., Meyer, J., and Kasper, G.: Calibration of an optical particle counter to provide mass for well-defined particle materials, Journal of Aerosol Science, 38, 325-332, 10.1016/j.jaerosci.2006.12.001, 2007.

Birmili, W., Rehn, J., Vogel, A., Boehlke, C., Weber, K., and Rasch, F.: Micro-scale variability of urban particle number and mass concentrations in Leipzig, Germany, Meteorologische Zeitschrift, 22, 155-165, 10.1127/0941-2948/2013/0394, 2013.

Brantley, H. L., Hagler, G. S. W., Kimbrough, E. S., Williams, R. W., Mukerjee, S., and Neas, L. M.: Mobile air monitoring data-processing strategies and effects on spatial air pollution trends, Atmospheric Measurement Techniques, 7, 2169-2183, 10.5194/amt-7-2169-2014, 2014.

Cai, J., Yan, B., Kinney, P. L., Perzanowski, M. S., Jung, K. H., Li, T., Xiu, G., Zhang, D., Olivo, C., Ross, J., Miller, R. L., and Chillrud, S. N.: Optimization approaches to ameliorate humidity and vibration related issues using the microAeth black
carbon monitor for personal exposure measurement, Aerosol Sci Technol, 47, 1196-1204, 10.1080/02786826.2013.829551, 2013.

Castell, N., Dauge, F. R., Schneider, P., Vogt, M., Lerner, U., Fishbain, B., Broday, D., and Bartonova, A.: Can commercial low-cost sensor platforms contribute to air quality monitoring and exposure estimates?, Environ Int, 99, 293-302, 10.1016/j.envint.2016.12.007, 2017.

Chien, C.-H., Theodore, A., Wu, C.-Y., Hsu, Y.-M., and Birky, B.: Upon correlating diameters measured by optical particle counters and aerodynamic particle sizers, Journal of Aerosol Science, 101, 77-85, 10.1016/j.jaerosci.2016.05.011, 2016.

Costabile, F., Alas, H., Aufderheide, M., Avino, P., Amato, F., Argentini, S., Barnaba, F., Berico, M., Bernardoni, V., Biondi, R., Casasanta, G., Ciampichetti, S., Calzolai, G., Canepari, S., Conidi, A., Cordelli, E., Di Ianni, A., Di Liberto, L., Facchini, M., Facci, A., Frasca, D., Gilardoni, S., Grollino, M., Gualtieri, M., Lucarelli, F., Malaguti, A., Manigrasso, M., Montagnoli,
M., Nava, S., Perrino, C., Padoan, E., Petenko, I., Querol, X., Simonetti, G., Tranfo, G., Ubertini, S., Valli, G., Valentini, S., Vecchi, R., Volpi, F., Weinhold, K., Wiedensohler, A., Zanini, G., Gobbi, G., and Petralia, E.: First Results of the "Carbonaceous Aerosol in Rome and Environs (CARE)" Experiment: Beyond Current Standards for PM10, Atmosphere, 8, 10.3390/atmos8120249, 2017.

Ghassoun, Y., Ruths, M., Lowner, M. O., and Weber, S.: Intra-urban variation of ultrafine particles as evaluated by process
related land use and pollutant driven regression modelling, Sci Total Environ, 536, 150-160, 10.1016/j.scitotenv.2015.07.051, 2015.

Ježek, I., Drinovec, L., Ferrero, L., Carriero, M., and Močnik, G.: Determination of car on-road black carbon and particle number emission factors and comparison between mobile and stationary measurements, Atmospheric Measurement Techniques, 8, 43-55, 10.5194/amt-8-43-2015, 2015.

Karjalainen, P., Pirjola, L., Heikkilä, J., Lähde, T., Tzamkiozis, T., Ntziachristos, L., Keskinen, J., and Rönkkö, T.: Exhaust particles of modern gasoline vehicles: A laboratory and an on-road study, Atmospheric Environment, 97, 262-270, 10.1016/j.atmosenv.2014.08.025, 2014.

Khlystov, A., Stanier, C., & Pandis, S. (2004). An Algorithm for Combining Electrical Mobility and Aerodynamic Size Distributions Data when Measuring Ambient Aerosol Special Issue ofAerosol Science and Technologyon Findings from the
Fine Particulate Matter Supersites Program. Aerosol Science And Technology, 38(sup1), 229-238. doi: 10.1080/02786820390229543

McMurry, P. H., Wang, X., Park, K., and Ehara, K.: The Relationship between Mass and Mobility for Atmospheric Particles: A New Technique for Measuring Particle Density, Aerosol Science and Technology, 36, 227-238, 10.1080/027868202753504083, 2002.

Müller, T., Henzing, J. S., de Leeuw, G., Wiedensohler, A., Alastuey, A., Angelov, H., Bizjak, M., Collaud Coen, M., Engström, J. E., Gruening, C., Hillamo, R., Hoffer, A., Imre, K., Ivanow, P., Jennings, G., Sun, J. Y., Kalivitis, N., Karlsson, H., Komppula, M., Laj, P., Li, S. M., Lunder, C., Marinoni, A., Martins dos Santos, S., Moerman, M., Nowak, A., Ogren, J. A., Petzold, A., Pichon, J. M., Rodriquez, S., Sharma, S., Sheridan, P. J., Teinilä, K., Tuch, T., Viana, M., Virkkula, A., Weingartner, E., Wilhelm, R., and Wang, Y. Q.: Characterization and intercomparison of aerosol absorption photometers:
result of two intercomparison workshops, Atmospheric Measurement Techniques, 4, 245-268, 10.5194/amt-4-245-2011, 2011.

Patton, A. P., Perkins, J., Zamore, W., Levy, J. I., Brugge, D., and Durant, J. L.: Spatial and temporal differences in traffic-related air pollution in three urban neighborhoods near an interstate highway, Atmos Environ (1994), 99, 309-321, 10.1016/j.atmosenv.2014.09.072, 2014.

Peters, J., Theunis, J., Van Poppel, M., and Berghmans, P.: Monitoring PM10 and Ultrafine Particles in Urban Environments
Using Mobile Measurements, Aerosol and Air Quality Research, 10.4209/aaqr.2012.06.0152, 2013.





Peters, J., Van den Bossche, J., Reggente, M., Van Poppel, M., De Baets, B., and Theunis, J.: Cyclist exposure to UFP and BC on urban routes in Antwerp, Belgium, Atmospheric Environment, 92, 31-43, 10.1016/j.atmosenv.2014.03.039, 2014.

Petzold, A., and Schönlinner, M.: Multi-angle absorption photometry—a new method for the measurement of aerosol light absorption and atmospheric black carbon, Journal of Aerosol Science, 35, 421-441, 10.1016/j.jaerosci.2003.09.005, 2004.

Petzold, A., Ogren, J. A., Fiebig, M., Laj, P., Li, S. M., Baltensperger, U., Holzer-Popp, T., Kinne, S., Pappalardo, G., Sugimoto, N., Wehrli, C., Wiedensohler, A., and Zhang, X. Y.: Recommendations for reporting "black carbon" measurements, Atmospheric Chemistry and Physics, 13, 8365-8379, 10.5194/acp-13-8365-2013, 2013.

Pfeifer, S., Müller, T., Weinhold, K., Zikova, N., Martins dos Santos, S., Marinoni, A., Bischof, O. F., Kykal, C., Ries, L., Meinhardt, F., Aalto, P., Mihalopoulos, N., and Wiedensohler, A.: Intercomparison of 15 aerodynamic particle size
spectrometers (APS 3321): uncertainties in particle sizing and number size distribution, Atmospheric Measurement Techniques, 9, 1545-1551, 10.5194/amt-9-1545-2016, 2016.

Rakowska, A., Wong, K. C., Townsend, T., Chan, K. L., Westerdahl, D., Ng, S., Močnik, G., Drinovec, L., and Ning, Z.: Impact of traffic volume and composition on the air quality and pedestrian exposure in urban street canyon, Atmospheric Environment, 98, 260-270, 10.1016/j.atmosenv.2014.08.073, 2014.

Rosenberg, P. D., Dean, A. R., Williams, P. I., Dorsey, J. R., Minikin, A., Pickering, M. A., and Petzold, A.: Particle sizing calibration with refractive index correction for light scattering optical particle counters and impacts upon PCASP and CDP data collected during the Fennec campaign, Atmospheric Measurement Techniques, 5, 1147-1163, 10.5194/amt-5-1147-2012, 2012.

Ruths, M., von Bismarck-Osten, C., and Weber, S.: Measuring and modelling the local-scale spatio-temporal variation of urban
particle number size distributions and black carbon, Atmospheric Environment, 96, 37-49, 10.1016/j.atmosenv.2014.07.020, 2014.

Van den Bossche, J., Peters, J., Verwaeren, J., Botteldooren, D., Theunis, J., and De Baets, B.: Mobile monitoring for mapping spatial variation in urban air quality: Development and validation of a methodology based on an extensive dataset, Atmospheric Environment, 105, 148-161, 10.1016/j.atmosenv.2015.01.017, 2015.

Van Poppel, M., Peters, J., and Bleux, N.: Methodology for setup and data processing of mobile air quality measurements to assess the spatial variability of concentrations in urban environments, Environ Pollut, 183, 224-233, 10.1016/j.envpol.2013.02.020, 2013.

Wiedensohler, A.: An Approximation of the Bipolar Charge Distribution for particles in the submicron size range, Journal of Aerosol Science, 19, 387389, 1988.

Wiedensohler, A., Birmili, W., Nowak, A., Sonntag, A., Weinhold, K., Merkel, M., Wehner, B., Tuch, T., Pfeifer, S., Fiebig, M., Fjäraa, A. M., Asmi, E., Sellegri, K., Depuy, R., Venzac, H., Villani, P., Laj, P., Aalto, P., Ogren, J. A., Swietlicki, E., Williams, P., Roldin, P., Quincey, P., Hüglin, C., Fierz-Schmidhauser, R., Gysel, M., Weingartner, E., Riccobono, F., Santos, S., Grüning, C., Faloon, K., Beddows, D., Harrison, R., Monahan, C., Jennings, S. G., O'Dowd, C. D., Marinoni, A., Horn, H. G., Keck, L., Jiang, J., Scheckman, J., McMurry, P. H., Deng, Z., Zhao, C. S., Moerman, M., Henzing, B., de Leeuw, G.,
Löschau, G., and Bastian, S.: Mobility particle size spectrometers: harmonization of technical standards and data structure to facilitate high quality long-term observations of atmospheric particle number size distributions, Atmospheric Measurement Techniques, 5, 657-685, 10.5194/amt-5-657-2012, 2012.

Wiedensohler, A., Wiesner, A., Weinhold, K., Birmili, W., Hermann, M., Merkel, M., Müller, T., Pfeifer, S., Schmidt, A., Tuch, T., Velarde, F., Quincey, P., Seeger, S., and Nowak, A.: Mobility particle size spectrometers: Calibration procedures
and measurement uncertainties, Aerosol Science and Technology, 52, 146-164, 10.1080/02786826.2017.1387229, 2018.

Williams, R. D., and Knibbs, L. D.: Daily personal exposure to black carbon: A pilot study, Atmospheric Environment, 132, 296-299, 10.1016/j.atmosenv.2016.03.023, 2016.

Yu, C. H., Fan, Z., Lioy, P. J., Baptista, A., Greenberg, M., and Laumbach, R. J.: A novel mobile monitoring approach to characterize spatial and temporal variation in traffic-related air pollutants in an urban community, Atmospheric Environment,
141, 161-173, 10.1016/j.atmosenv.2016.06.044, 2016.





Appendix A

**Mobile Measurement platform**

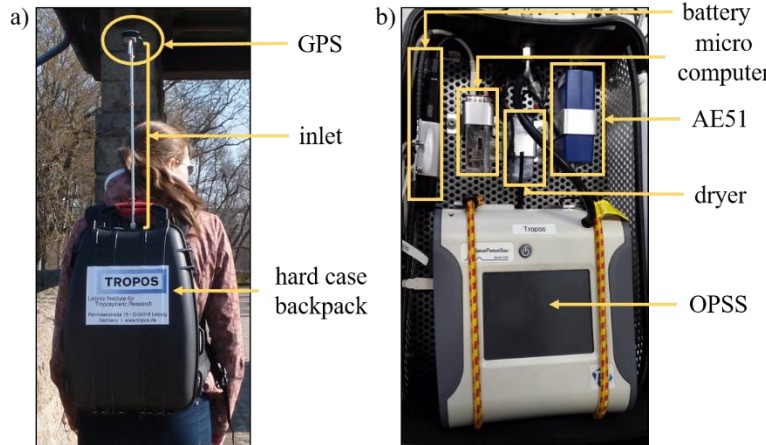

**Figure A 1 The TROPOS aerosol backpack from outside (a) showing the GPS unit, inlet, and entire backpack; and from inside (b) showing the different components and main instruments: AE51 and OPSS. Image source: TROPOS, 2016**

The mobile aerosol measurements were performed using a mobile platform (the "aerosol backpack") that was designed to be easily carried by a pedestrian as shown in Figure A1. A 1 m long stainless steel inlet protrudes from the top of the backpack

with a GPS device resting on a metallic plate on top. Ambient air is sampled through this inlet and then splits into two channels: one leading to an absorption photometer to measure eBC after passing a silica-based aerosol diffusion dryer, and other leading to the optical particle size spectrometer. The absorption photometer used in this study is the AethLabs microAeth® Model AE51. It determines eBC mass concentrations based on the attenuation of light (880 nm) passing through a particle-loaded filter and assuming a mass attenuation cross section (12.5 $m^2$ $g^{-1}$). The instrument operates on a time base of 1 s and a flow

rate of 100 mL $min^{-1}$. Since the AE51 has been known to be sensitive to sudden changes in humidity, the aerosol first passes through a dryer which is a Perma Pure membrane within an aluminium mesh. This mesh is then surrounded by silica gel granules. This set up is housed in a small sealed box to keep the silica gel granules dry and avoid leaks. The optical particle size spectrometer employed here is the TSI OPSS Model 3330 which provides an optical particle number size distribution (PNSD) from 0.3 µm to 10 µm divided into 16 channels. The aerosol enters the instrument with a flow of 1 L $min^{-1}$ and is led



into a detection chamber, where it crosses a vertically polarized laser beam with a wavelength of 660 nm. The light scattered by the particle is focused by a 120° spherical collecting mirror to the photodetector (more details in the instruments user manual). The intensity of the light pulse and counting rate are used to size and count and the particles, respectively. The aerosol sampled by the OPSS is not dried which may influence the measurements in humid environments. The data acquisition

is controlled and synchronized by a microcomputer which is powered by a battery package. The same battery package powers the AE51 and GPS while the OPSS has its own source. All these instruments (except the GPS) are secured inside a waterproof, hard case backpack.

Appendix B

**Fixed reference station**

A fixed reference measurement station was set up within a garden with restricted traffic and public access and was approximately 100–400 m away from the major roads. The station is a mobile laboratory (AEROLAB), designed for ambient aerosol and gas measurements with controlled indoor conditions. The sampling system consisted of a $PM_{10}$ inlet followed by

an aerosol diffusion dryer, conditioning the relative humidity level below 40 %. The aerosol then passed through an isokinetic splitter, distributing the aerosol to the different instruments, including the calibrated and quality-assured reference instruments used for this study: a MAAP for eBC (multi-angle absorption photometer, Model 2012, Thermo, Inc., Waltham, MA USA), a TROPOS-type MPSS for the PNSD (Wiedensohler et al. (2012)) and an APSS (aerodynamic particle size spectrometer, Model 3321 TSI Inc., Shoreview, MN USA).

MAAP determines the aerosol light absorption coefficient at a wavelength of 637 nm (Petzold and Schönlinner, 2004) with one-minute time resolution. The mass concentration of eBC is then internally calculated, using a mass absorption cross section of 6.6 $m^2$ $g^{-1}$. The MPSS classifies electrical particle mobility in the range 0.01 to 0.80 µm (mobility diameter, $D_{p,Mob}$). Using the standardized bipolar charge distribution, the PNSD can be calculated (Wiedensohler, 1988). The uncertainty of the reference instrument in terms of the integral number concentration is smaller than 10 % and in terms of sizing is smaller than

3 % (Wiedensohler et al.,2018). The APSS determines the PNSD using the time of flight in an accelerated flow to determine the aerodynamic particle diameter. The aerodynamic particle size range covered by the APSS is approximately 0.7 to 10 µm for atmospheric applications. Both the MPSS and the APSS were operated with a time resolution of 5 minutes.