# Peer review of "Methodology for High Quality Mobile Measurement with Focus on Black Carbon and Particle Mass Concentrations"

_Atmospheric Measurement Techniques, 2019_

## Referee Comment (RC1) · Anonymous Referee #2 · 3 May 2019

The manuscript 'Methodology for High Quality Mobile Measurement with Focus on Black Carbon and Particle Mass Concentrations" by Alas et al. describe a set of steps to assure the best quality of mobile aerosol measurements. Overall, the study is interesting and I think the information provided is very useful to the aerosol community. I would recommend acceptance of this paper with minor revisions.

GENERAL COMMENTS 1. It would be interesting if an estimate of errors would be provided "simulating" cases in which some of the steps suggested are not followed. This is because it is likely that during future campaigns similar to this all the instrumentation necessary for the detailed comparison might not be available or within the budget of

a project. For example, what would be the error introduced if one would not have the option of duplicate runs? This could be easily quantified with the data at hand. 2. The paper is mostly well written but there are several instances where verb-subject number agreement should be corrected (some examples in the specific comments section)

SPECIFIC COMMENTS Abstract, Page 1, Line 18: "can provide following" should be "can provide the following" Page 1, line 23: "physical meaningful" probably should be "physically meaningful" Page 1, line 26: "MPSS+APSS" should be "MPSS+OPSS"? Table 1: I wish they had also used photoacoustic or extinction minus scattering techniques to check for accuracy (not just precision) Section 3.1.1: This is good, but on what particles will the aethalometer and MAAP comparison be carried out? Page 8, line 10: It would be good to provide the cause of the underestimation. Page 8, line 13: Again, it would be nice to know the potential reason. Page 8, line 19-20: "The scatter plots on the right of each time series shows..." should be "The scatter plots on the right of each time series show" because the verb refers to plots (plural) Page 8, line 25: "Large differences, on the other hand, were investigated further to determine if it is related..." should read "Large differences, on the other hand, were investigated further to determine if they are related" because the subject is "Large differences" Page 10: I had a little bit of a hard time to follow the section on "Convergence Analysis" Page 11, line 11: "the data points has to be spatially" should be "the data points have to be spatially" Page 11, line 12: "data points that is not part of the route" should be "data points that are not part of the route" Section 3.2.2: I would have liked some more guidelines on criteria to select background sites. Page 13, line 25: "measurements which uses" should be "measurements which use" Page 13, line 28: "shape factor = 1" how good is this assumption? Page 14, line 18: "The agreement for PM10 is lower" please quantify. Page 15, line 12: "AE51 data was" should probably be "AE51 data were" Page 15, line 14: it should be "mass concentration is" or "mass concentrations are" Page 21, line 7: either "unique CFf,vol are applied" or "a unique CFf,vol is applied"
AMTD

---

## Author Comment (AC1) · 13 May 2019

Interactive comments on "Methodology for High Quality Mobile Measurement with Focus on Black Carbon and Particle Mass Concentrations" by Alas et al.

Referee comments are noted in black and denoted with "RC". Author replies/comments are in blue and denoted with "AC". Changes in the manuscript are in blue as well, italicized, and denoted with "Change in Manuscript"

We would like to thank the Referee for recommending this manuscript for acceptance and for the constructive comments. Please find our response to each of the comments below. Attached is the revised version of the manuscript with the changes marked.

**Anonymous Referee #2**

GENERAL COMMENTS:

RC: 1. It would be interesting if an estimate of errors would be provided "simulating" cases in which some of the steps suggested are not followed. This is because it is likely that during future campaigns similar to this all the instrumentation necessary for the detailed comparison might not be available or within the budget of a project. For example, what would be the error introduced if one would not have the option of duplicate runs? This could be easily quantified with the data at hand.

AC: Thank you for your comment. We understand the need to estimate errors for cases when the methodology proposed is not feasible. For the most part, specifically the quality checks in laboratory and field, an infinite number of technical errors can occur which would be difficult to estimate. One of the main goals of the method proposed is to minimize, detect, and address these errors as soon as possible. Although, we do agree that we can estimate the errors that we experienced for the following parts, especially the ones we have experienced ourselves:

> Parallel runs → without another unit when the instruments had unflagged technical errors or drifts, and in the absence of live viewing of the data, eBC mass concentrations and particle number concentrations had errors of 50% and 19-80%, respectively. We have mentioned this in the text but not quantified (Page 8 line 12 to 16).
> Refractive index and fine mode volume corrections for the OPSS data → The impact of these corrections on the size distribution derived PM2.5 from the OPSS, we believe, is discussed comprehensively and quantified in Figure 13, page 22.

Change in Manuscript: Page 8 lines 12-16

*"For example, during the early stages of the campaign, analysis of the collocated measurements revealed that one AE51 was underestimating eBC mass concentrations by 50% due to weakening of the pump causing the flow to decrease. This was not flagged by the instrument, but because another AE51 was in operation, the error was identified and corrected immediately. Similarly, towards the end of the campaign, due to unidentified reasons, the sheath flow of one of the OPSS started to increase which resulted to an underestimation of the particle number concentration (PNC) across all size bins (19 – 80%)."*

RC: 2. The paper is mostly well written but there are several instances where verb-subject number agreement should be corrected (some examples in the specific comments section)

AC: Thank you for catching our grammatical mistakes. We have improved the manuscript in this regard and the changes are written below following your "specific comments".

SPECIFIC COMMENTS:

RC: Abstract, Page 1, Line 18: "can provide following" should be "can provide the following"

AC: Changed.

Changes: Page 1, Line 18: "*The application of the methodology can provide the following results.*"

RC: Page 1, line 23: "physical meaningful" probably should be "physically meaningful"

AC: Changed.

Changes: Page 1, Line 23: "*...distribution using physically meaningful corrections.*"

RC: Page 1, line 26: "MPSS+APSS" should be "MPSS+OPSS"?

AC: In this sentence, we are referring to the "reference instrument" which is the combination of the MPSS and APSS (mobility and aerodynamic particle size spectrometers). For clarity, the following changes were made on the sentence involved:

Changes: Page 1, Line 24-26: "*Using size-resolved complex refractive indices and time-resolved fine mode volume correction factors of the fine particle range, the calculated $PM_{2.5}$ from the OPSS was within 5 % of the reference instruments (MPSS+APSS).*"

RC: Table 1: I wish they had also used photoacoustic or extinction minus scattering techniques to check for accuracy (not just precision).

AC: The MAAP and AE33 in the fixed station were both regularly quality-assured at the World Calibration Centre for Aerosol Physics in TROPOS, Germany. The instruments are quality-assured through intercomparisons of different filter-based instruments using generated pure black carbon particles of known mass absorption coefficients. This procedure is detailed in Müller et al., 2011, AMT.

Changes: No changes were made.

RC: Section 3.1.1: This is good, but on what particles will the aethalometer and MAAP comparison be carried out?

AC: For this case, to capture the performance of the AE51 in real world scenarios, the intercomparison between the MAAP and AE51 were carried out through parallel long-term measurements (overnight or over multiple days) of ambient air from the same inlet. The same procedure is done during in-field quality assurance as described in this study.

Changes: Page 6 Line 18-20 (Section 3.1.1) *"The AE51 units must be compared against a multi-angle absorption photometer, (MAAP Model 2012, Thermo, Inc., Waltham, MA USA), provided that both are connected to the same inlet, with ambient air to test the performance of the AE51 in real-world scenarios."*

RC: Page 8, line 10: It would be good to provide the cause of the underestimation.

AC: The cause of the underestimation of one of the AE51 units was due to the weakening of the pump. The flow was decreasing and needed to be recalibrated which we did.

Changes: Page 8, Line 12-13 (previously line 10): *"For example, during the early stages of the campaign, analysis of the collocated measurements revealed that one AE51 was underestimating eBC mass concentrations by 50% due to weakening of the pump causing the flow to decrease."*

RC: Page 8, line 13: Again, it would be nice to know the potential reason.

AC: Unfortunately, we never figured out why the sheath flow of one of the OPSS drifted during the campaign. We brought up the issue with the manufacturer but received no conclusive answer.

Changes: Page 8 Line 15-16 (previously line 13): "*Similarly, towards the end of the campaign, due unidentified reasons, the sheath flow of one of the OPSS started to increase which resulted to an underestimation of the particle number concentration (PNC) across all size bins (19 – 80%).*"

RC: Page 8, line 19-20: "The scatter plots on the right of each time series shows. . ." should be "The scatter plots on the right of each time series show" because the verb refers to plots (plural)

AC: Changed.

Changes: Page 8, Line 22-23 (previously line 19-20): *"The scatter plots on the right of each time series show the correlation between the two corresponding instruments."*

RC: Page 8, line 25: "Large differences, on the other hand, were investigated further to determine if it is related. . ." should read "Large differences, on the other hand, were investigated further to determine if they are related" because the subject is "Large differences"

AC: Changed.

Changes: Page 8, line 28-29 (previously line 25-26): *"Large differences, on the other hand, were investigated further to determine if they are related to sources or technical malfunctions."*

RC: Page 10: I had a little bit of a hard time to follow the section on "Convergence Analysis"

AC: My apologies. The text has been modified.

Changes: Page 10, Convergence Analysis section: *"The idea is to take the pollutant concentrations measured per run along a specific part of the route. Then take the cumulative (increasing number of runs) average (or median) of those concentrations. This procedure is done with high number of iterations to achieve high number of possible combinations of the runs. Convergence is achieved when the iterations has stabilized to an asymptotic behavior towards the desired metric (e.g. median concentration from that location from all runs). The number of runs when the iterations are within the specified threshold of deviation from the selected metric (criteria for convergence) then tells how many runs are needed to achieve the representative concentration."*

RC: Page 11, line 11: "the data points has to be spatially" should be "the data points have to be spatially"

AC: Changed.

Changes: Page 11, Line 11: *"Therefore, to obtain the overall spatial distribution, the data points have to be spatially aggregated."*

RC: Page 11, line 12: "data points that is not part of the route" should be "data points that are not part of the route"

AC: Changed

Changes: Page 11, line 12: *"Prior to spatial aggregation, the data cloud has to be cleaned by removing data points that are not part of the route (e.g. detours, inaccurate GPS points)."*

RC: Section 3.2.2: I would have liked some more guidelines on criteria to select background sites.

AC: We agree that more information regarding the selection of the background site is needed. We referred to the guidelines provided in the Air Quality Directive 2008/50/EC:
"Urban background locations shall be located so that their pollution level is influenced by the integrated contribution from all sources upwind of the station. The pollution level should not be dominated by a single source unless such a situation is typical for a larger urban area. Those sampling points shall, as a general rule, be representative for several square kilometers"

Changes: Page 13, Section 3.2.2: *"The fixed station containing reference instruments is crucial for the quality assurance of the mobile instruments and also for the determination of $PM_{2.5}$ mass concentration derived from the PNSD of the OPSS mobile measurements. Therefore, the selection of the fixed measurement site should be taken with care. An urban background location should be selected as fixed station, namely (as stated in the 2008/50/EC Air Quality Directive) a site located in an area that is not dominated by a single source and instead captures the combination of all the sources upwind of the selected site. For this study, the following criteria were followed: 1) the site should be inaccessible or has limited accessibility to vehicles; 2) the site should not be <100 m away from any main thoroughfare; 3) there should be minimal obstruction (e.g. buildings) in its immediate vicinity. The decision on the reference site location is also a balance between scientific aims and availability of space. Since this study was conducted in the city of Rome, the fixed station was placed inside a government-owned garden that is inaccessible to*

*most non-government vehicles and is 115 m away from the nearest trafficked road. The site can be considered representative of the fine particulate matter at urban background locations in Rome as its average values of PM$_{2.5}$ mass concentrations are consistent with typical values measured at the urban background sites of the local air quality monitoring network (cf. Table 4 in Costabile et al. 2017)."*

RC: Page 13, line 25: "measurements which uses" should be "measurements which use"

AC: Changed.

Changes: Page 13, line 25: *"…which is an advantage over filter-based measurements which use gravimetric analyses to obtain PM mass concentrations."*

RC: Page 13, line 28: "shape factor = 1" how good is this assumption?

AC: At the CARE urban background site, a large fraction of particles is expected to be aged in the atmosphere. This fraction is composed of submicrometer aged particles with diameters larger than approx.100 nm, and is supposed to be spherical. Supermicrometer particles (e.g., dust) might be not spherical, but were excluded from the analysis.

Particles smaller than 100 nm might include shortly aged soot particles. The shape of these particles (usually fractal when freshly emitted) changes with their aging in the urban atmosphere. They can experience a conversion from fresh fractal to aged spherical shapes by becoming more compact and the overall particle spherical when coated in other inorganic and organic material. The shortly aged soot particles are supposed to be a fraction of the total BC (including aged biomass burning particles, as well). Therefore, we expect that the shortly aged soot particles account for less than approx. 10% of the PM$_{10}$ (i.e., average value estimated for BC/PM$_{10}$). A smaller fraction of these shortly aged soot particles might be not spherical, but cannot be exactly quantified here.

Furthermore, PM$_{2.5}$ and PM$_{10}$ mass concentrations derived from this procedure with this assumption compared well with those measured by a beta-attenuation monitor at another urban background site in Rome as stated in the next paragraph. Although, we must say, the assumption that the particles are spherical could have also contributed to the lower correlation of PM$_{10}$ as larger particles becomes more irregular in shape with increasing size.

Changes: Page 14, line 6 – 7 (previously page 13 line 28): *"The conversion assumed aged spherical particles in the fine mode (shape factor = 1) as expected at urban background regions, and a size-dependent particle density (1.6 to 2 g cm-3)."*

RC: Page 14, line 18: "The agreement for PM$_{10}$ is lower" please quantify.

AC: Changed.

Changes: Page 14, line 18: "The agreement for PM$_{10}$ is lower (r2 = 0.73, y = 0.88x), probably because of dust and marine aerosol events, which are supposed to modify the particle density used in the calculation of particle mass from particle number size distributions …"

RC: Page 15, line 12: "AE51 data was" should probably be "AE51 data were"

AC: Changed.

Changes: Page 15, line 12: *"For the eBC mass concentration measurements, the AE51 data were averaged per minute to compare with the MAAP."*

RC: Page 15, line 14: it should be "mass concentration is" or "mass concentrations are"

AC: Changed.

Changes: Page 15, Line 14: *"This increases our confidence that the measured eBC mass concentrations are reliable for the entire route."*

RC: Page 21, line 7: either "unique CFf,vol are applied" or "a unique CFf,vol is applied"

AC: Changed.

Changes: Page 21, line 7: *"Finally, the deviation from the reference is significantly minimized when a size-resolved refractive index correction is used and a unique CFf,vol is applied for each run."*

---

## Author Comment (AC2) · 13 May 2019

Please find the revised manuscript with tracked changes as a supplement file.

Please also note the supplement to this comment:
https://www.atmos-meas-tech-discuss.net/amt-2019-66/amt-2019-66-AC2-supplement.pdf

---

## Referee Comment (RC2) · Anonymous Referee #1 · 14 Jun 2019

This manuscript provides recommended techniques and protocols for mobile measurements of particulate matter. The recommendations stem largely from planning and implementation of the 2017 CARE study in Rome. As the authors note, the utility of mobile measurements has increased in recent years with increased availability of more compact (and less costly) instruments and with growing appreciation of the extent and importance of heterogeneity in air quality and its relevance for exposure and health impacts. It seems appropriate and valuable to have some guidelines for the community that can lead to improved data quality and inter-study comparability. A challenge in developing such guidelines is that the techniques, tools, and goals of different studies are diverse. Some of the recommendations provided in the manuscript are sufficiently

broad to be more generally useful, but many are too restrictive and would be applicable only for studies very similar to the CARE study. Furthermore, much of the guidance is pretty straightforward and would be considered common sense for many readers, while some such as the OPSS correction procedure, though useful, seems outside the primary focus of the paper. I agree that the analysis and correction techniques used for the specific set of instruments used during CARE are valuable, but it seems they should accompany the discussion of the results from that study (presumably in Costabile et al., 2017) and not be included here. Portions of the manuscript were seemingly adapted from step-by-step protocols or best-practices employed by the research team. Though there is nothing inherently wrong with doing so, the results is certain recommendations that aren't needed for the readers of AMT such as the importance of calibration (the subject of three sub-sections), and others that need to be modified to be relevant for researchers using different instruments in different environments and with different objectives. I feel that this could be publishable, but only following major revision to shift the emphasis towards the more general guidelines for making and analyzing mobile measurements. It would also be valuable for the authors to discuss what they might do (or already have done) differently based on what was learned during the CARE study. Are there alternative instruments or techniques that they are considering? And how would recommendations differ for measurements made with a CPC and/or a filter pack, with or without an OPSS?

Minor comments in order of the location in the text:

Page 3, line 10: I don't question that the measurements were well done and the dataset was valuable, but these don't seem to me to be elaborate.

Table 1: The authors should discuss the tradeoffs between using an arguably more accurate reference instrument (e.g., MAAP) and a duplicate of that used for the mobile measurements. I appreciate that there are advantages, but issues such as different wavelengths and potentially differing interferences from scattering particles, humidity, . . . introduce uncertainty.

Table 1: The TSI OPSS model number should be provided here (I recognize that it is provided in the text).

Page 6, line 18: Related to my comments above, statements such as "The AE51 units must be compared against a multi-angle absorption photometer (MAAP…" may have been useful for the authors during their study but would not be for groups using other combinations of instruments.

Page 6, line 25: And related to other comments made above, statements such as "In deciding on the length of the length of the route and the duration of a run…the operating time of the instruments and rest time for the runners should be considered. If multiple runs are done within one day, the charging time of the instruments should be considered as well." simply seems too evident to include in a scientific manuscript.

Page 9, line 5: This is a more general comment, but is most closely related to the discussion starting here. Some consideration should be given to the potential bias introduced by following the same route each day while emissions and meteorology change in a somewhat predictable way. The use of the reference site may help account for concentration trends caused by factors such as boundary layer height development in the morning. But the choice to put the reference site away from the largest emissions sources could result in greater sensitivity to boundary layer dynamics along the route close to sources than at the reference site.

Page 18, line 1: Related to the comment above about things the authors might do differently next time, it would be useful to include a discussion here about what tradeoffs they feel would be justified to have an OPSS capable of detecting smaller particles. It simply seems that the uncertainty introduced by the corrections needed could be reduced significantly.

---

## Author Comment (AC3) · 10 Jul 2019

Interactive comments on "Methodology for High Quality Mobile Measurement with Focus on Black Carbon and Particle Mass Concentrations" by Alas et al.

Referee comments are noted in black and denoted with "RC". Author replies/comments are in blue and denoted with "AC". Changes in the manuscript are in blue as well, italicized, and denoted with "Change in Manuscript"

We would like to thank the Referee for the constructive comments. Please find our response to each of the comments below. Attached is the revised version of the manuscript with the changes marked.

**Anonymous Referee #1**

**GENERAL COMMENTS:**

RC: A challenge in developing such guidelines is that the techniques, tools, and goals of different studies are diverse. Some of the recommendations provided in the manuscript are sufficiently broad to be more generally useful, but many are too restrictive and would be applicable only for studies very similar to the CARE study. Furthermore, much of the guidance is pretty straightforward and would be considered common sense for many readers, while some such as the OPSS correction procedure, though useful, seems outside the primary focus of the paper. I agree that the analysis and correction techniques used for the specific set of instruments used during CARE are valuable, but it seems they should accompany the discussion of the results from that study (presumably in Costabile et al., 2017) and not be included here. Portions of the manuscript were seemingly adapted from step-by-step protocols or best-practices employed by the research team. Though there is nothing inherently wrong with doing so, the results is certain recommendations that aren't needed for the readers of AMT such as the importance of calibration (the subject of three sub-sections), and others that need to be modified to be relevant for researchers using different instruments in different environments and with different objectives. I feel that this could be publishable, but only following major revision to shift the emphasis towards the more general guidelines for making and analyzing mobile measurements. It would also be valuable for the authors to discuss what they might do (or already have done) differently based on what was learned during the CARE study. Are there alternative instruments or techniques that they are considering? And how would recommendations differ for measurements made with a CPC and/or a filter pack, with or without an OPSS?

AC: Thank you for your comments. We would like to answer your general comments point by point.

Our apologies if we were not able to make it clear in our manuscript that the main goal of this paper is to present a methodology that ensures high quality data of eBC and PM2.5 mass concentrations from mobile measurements for scientific purposes by, among other things, having measurements that are traceable through the site-intercomparisons against calibrated reference instruments. We would like to point out as well, that the CARE data was used as an example and this method can be applied in a more general way. For instance, regardless of the instruments or objectives of a particular study, one may still follow the methods proposed here to achieve high quality data (calibration and checks of the instruments before, during, and after deployment; intercomparison between mobile and reference instrument, collocated measurements, and so on). The more restrictive parts of this method are due to the current portable instrumentation available for eBC mass concentrations and particle number size distribution. To achieve high quality data, we focused our method on instruments for eBC and particle number size distribution that are well characterized and widely used. We do recognize that studies with different goals would use different instruments

and techniques. In Section 3.2.4 (page 15), we mentioned other options for particle mass measurements that are traceable to SI units such as online monitors, filter-based measurements, and chemical analyses. To address this, we improved the current manuscript to make our goals clearer for the readers that these methods can applied to any mobile measurement experiment with a fixed site that contains reference instruments and that we used the CARE data as an example to demonstrate these methods.

- In the creation of this manuscript, we decided to be as thorough as possible and included information that are common practice in our community. The reason behind is we want to be as informative and educative as possible to readers who are new in the field whom we have encountered often enough during our campaigns in different parts of the world, particularly in developing regions. In addition, some parts such as the calibration of instruments in the fixed station, are here and not in Costabile et al., 2017 because that paper is an overview paper of the whole campaign involving numerous institutions with different scientific question, instruments, and techniques. Whereas this study focuses on the technical aspect of the mobile measurements which involves the quality-assurance of the reference instruments as well.
- The OPSS correction procedure is one of the main focus of this manuscript. As we are going for quality-assured measurements of PM that can be used for scientific purposes, we opted for measurements of particle number size distribution that we can convert to particle mass (conversion procedure that is traceable) using physically meaningful assumptions and corrections and known uncertainties. We did not want to use PM sensors that give out PM mass concentrations without full knowledge on how the numbers come about. Hence, we would like to share this knowledge to our readers who aim to have quality-assured PM mass measurements from mobile platforms and not only indicative values.
- The majority of the methodology proposed here, we believe, will remain the same regardless of changes in instrumentation. For example, an OPSS with lower detection limit may reduce the uncertainty of the overlap with the MPSS in the fine mode, but the method to assure its quality will remain the same (frequent field intercomparisons, refractive index correction, fine mode volume correction). The same is the case when a newer absorption photometer is used with several wavelengths. More information about the ambient aerosol can be acquired, but the method to assure its quality will not change. If there will be a change in instrumentation (i.e. CPC), the methods presented here may still be followed (the mobile CPC should still be calibrated and checked, compared against a reference CPC, and compared against each other through collocated measurements). But that is not within the scope of this manuscript since we can't calculate PM from total particle number concentrations.

Changes in manuscript:

• Page 3, line 8-12: "The main goal of this article is to propose a methodology for mobile measurements and data processing, which would provide reliable and quality-assured data of spatially resolved eBC and PM mass concentrations for scientific purposes. Specifically, we propose measurements and post-processing techniques based on meaningful physical assumptions

by addressing the limitations of an OPSS. Measurements from an intensive campaign in Rome, Italy was used to demonstrate the proposed methodology."

- Page 6, line 1: "Exemplary measurements"
- Page 6 line 2-6: "To demonstrate the proposed methodology, we used the measurements from the mobile measurement experiment that was part of an intensive campaign called Carbonaceous Aerosols in Rome and Environs (CARE) in the downtown area of Rome, Italy, in February of 2017. The scientific aim of CARE was to characterize the carbonaceous aerosol in the Mediterranean urban background area of Rome. An overview of this campaign and the first results are presented by Costabile et al. (2017)."
- Page 7, line 6-12: "Each mobile measurement period should include a pre-run routine: 1) checking for leaks within the systems by placing a total filter on the inlet, 2) giving ample time for the instruments to warm up (depending on the instruments used), 3) measuring the total flow of the system, and 4) synchronizing the time of the two microcomputers or data loggers of each backpack. Additionally, if the pre-run routine is done indoors, then once stepping outside, the GPS should be given enough time to fine satellites to get accurate location data before starting the run. Other instrument-specific routines should also be included. For instance, in this study, the filter of the AE51 was replaced before each run to avoid filter saturation."
- Page 14, lines 2-3 (Section 3.2.4): "As one of the main objectives of this study is to provide a methodology for high quality measurements of PM, this subsection goes into detail of how this is achieved when calculating PM from PNSDs."
- Pahe 15, lines 12 19 (Section 3.3): "Having a fixed site with reference instruments provide the opportunity to check the performance of the mobile instruments in the field relative to the day-today changes (i.e. emissions, meteorology) within the study area. Performing sufficiently long and frequent intercomparisons against the reference instruments in the middle of a run further ensures the quality of the data from the mobile measurements. Furthermore, the intercomparisons harmonizes the OPSS and MPSS+APSS at the reference site which allows for the correction the OPSS PNSD per run based on the relative changes occurring in the study area. In this study, the runners stop by the fixed station for 30 minutes in the middle of each run for intercomparisons against the reference instruments."
- Page 16, line 5: "This section provides a detailed and traceable method of calculating PM from the OPSS PNSD."
- Page 23, line 9-10 (Conclusions): "A methodology to assure high quality mobile measurement data of eBC and PM2.5 mass concentrations was introduced and demonstrated using exemplary measurements from an intensive field study in Rome, Italy, February 2017."

**SPECIFIC COMMENTS:**

RC: Page 3, line 10: I don't question that the measurements were well done and the dataset was valuable, but these don't seem to me to be elaborate

AC: Changed: Word "elaborate" deleted.

Change in Manuscript: Page 3, Line 9-11: "Specifically, we propose measurements and post-processing techniques based on meaningful physical assumptions by addressing the limitations of an OPSS."

RC: Table 1: The authors should discuss the tradeoffs between using an arguably more accurate reference instrument (e.g., MAAP) and a duplicate of that used for the mobile measurements. I appreciate that there are advantages, but issues such as different wavelengths and potentially differing interferences from scattering particles, humidity, . . . introduce uncertainty.

AC: Thank you, and we do understand the value of this. In section 3.2.1 we cited Muller et al., 2011, which presented the results of intercomparisons of different absorption photometers. We opted to not include these details in this manuscript as Muller et al., 2011 have fully characterized, compared, and analyzed these instruments already. Furthermore, the MAAP is what was used in the CARE campaign. We believe the tradeoffs will not be significant if other absorption photometers were used (i.e. AE33) as long as the correlation between the chosen reference absorption photometer and the mobile absorption photometer is known beforehand (i.e. laboratory experiments and intercomparisons). The main point here is that the mobile instruments are harmonized at the fixed site with the reference instrument.

Change in Manuscript: None

RC: Table 1: The TSI OPSS model number should be provided here (I recognize that it is provided in the text).

AC: The latest version of the manuscript (the one made addressing the comments of Referee #2) has the model number of the OPSS.

RC: Page 6, line 18: Related to my comments above, statements such as "The AE51 units must be compared against a multi-angle absorption photometer (MAAP. . . ." may have been useful for the authors during their study but would not be for groups using other combinations of instruments.

**AC: Changed.**

Change in Manuscript: Page 6, Lines 18-21: "The AE51 units must be compared against a well characterized and calibrated optical absorption photometer, in this case a multi-angle absorption photometer (MAAP Model 2012, Thermo, Inc., Waltham, MA USA), provided that both are connected to the same inlet with ambient air to test the performance of the AE51 in real-world scenarios."

RC: Page 6, line 25: And related to other comments made above, statements such as "In deciding on the length of the length of the route and the duration of a run. . .the operating time of the instruments and rest time for the runners should be considered. If multiple runs are done within one day, the charging time of

the instruments should be considered as well." simply seems too evident to include in a scientific manuscript.

AC: Changed. Lines 26-29 were deleted.

Change in Manuscript: -

RC: Page 9 Line 5: This is a more general comment, but is most closely related to the discussion starting here. Some consideration should be given to the potential bias introduced by following the same route each day while emissions and meteorology change in a somewhat predictable way. The use of the reference site may help account for concentration trends caused by factors such as boundary layer height development in the morning. But the choice to put the reference site away from the largest emissions sources could result in greater sensitivity to boundary layer dynamics along the route close to sources than at the reference site.

AC: We measure along a fixed route repeatedly to achieve representativeness. As mentioned in the convergence analysis part, single runs along a particular street may not give us concentrations that are representative of that area since it will be sensitive to impacts of single events. Furthermore, in this study, we are focused on determining the spatial variability of eBC and PM in different microenvironments. By running repeatedly and with high frequency, we cancel out the influence of the boundary layer and other larger meteorological phenomena.

If you are pertaining to the variability of the particle volume size distribution (our OPSS fine mode volume correction assumes that the PVSD in the urban background area is similar elsewhere), we have addressed this with a data experiment from other stations in Germany in Section 3.3.3 (page 19-20).

"This assumption comes with limitations as it doesn't account for the likely differences of the aerosol sources along the entire route. A data experiment was performed comparing PVSDs obtained at an urban background station and at a roadside station in the city of Dresden, Germany for the whole month of February 2017. For each site, effective correction factors (CFf,vol) were calculated for each hour between 6 AM and 9 PM using the fraction covered by the OPSS as a proxy for the OPSS size distribution. For background station CFf,vol < 2, which represented ~50 % of the hourly data, there was excellent agreement between the paired background and roadside CFf,vol values with 3 % mean bias and narrow variability (1 sigma = 5 %). For higher CFf,vol values the bias increased with increasing CFf,vol and approached 20% for CFf,vol > 3 (background station having higher values and the variability also increased (1 sigma =11 %). While caution must be used in extrapolating the Dresden data to other locations and conditions, these results provide a context for understanding the limitations when using a correction factor derived at a single location to represent the behaviour along the entire route. For most cases, the impacts of location-dependent CFf,vol values will be damped through the use of repeat runs although some concentration bias might remain."

Change in Manuscript: Page 10, lines 1-2: *"Furthermore, frequent runs will also average out the influence of meteorology such as dynamics of the boundary layer and different wind conditions."*

RC: Pag 18 Line 1: Related to the comment above about things the authors might do differently next time, it would be useful to include a discussion here about what tradeoffs they feel would be justified to have an OPSS capable of detecting smaller particles. It simply seems that the uncertainty introduced by the corrections needed could be reduced significantly

AC: We appreciate this comment. In the past, we have had experience comparing different portable optical size spectrometers available in the market and the TSI OPSS 3330 proved to be the most reliable based on laboratory experiments and comparability with an aerodynamic particle size spectrometer. However, we believe that having an OPSS with lower detection limit may reduce the uncertainty between the overlap with the MPSS but not significantly. The method necessary to achieve high quality data from an instrument with lower detection limit will remain the same.

Change in Manuscript: None.

**Methodology for High Quality Mobile Measurement with Focus on Black Carbon and Particle Mass Concentrations**

Honey Dawn C. Alas1, Kay Weinhold1, Francesca Costabile2, Antonio Di Ianni2, Thomas Müller1, Sascha Pfeifer1, Luca Di Liberto2, Jay R. Turner3, and Alfred Wiedensohler1

[revised manuscript text omitted]